# Simultaneous organic aerosol source apportionment at two Antarctic sites reveals large-scale and eco-region specific components

Marco Paglione[1], David C.S. Beddows[2], Anna Jones[3], Thomas Lachlan-Cope[3] Matteo Rinaldi[1], Stefano Decesari[1], Francesco Manarini[1], Mara Russo[1], Karam Mansour[1*], Roy M. Harrison[2**], Andrea Mazzanti[4], Emilio Tagliavini[4], Manuel Dall'Osto[5]

[1]Italian National Research Council - Institute of Atmospheric Sciences and Climate (CNR-ISAC), Bologna, 40129 Italy
[2]National Centre for Atmospheric Science, University of Birmingham, Edgbaston, Birmingham, B15 2TT, United Kingdom
[3]British Antarctic Survey, NERC, Cambridge, CB3 0ET, United Kingdom
[4]Department of Chemistry, University of Bologna, Bologna, 40126, Italy
[5]Institute of Marine Sciences, CSIC, Barcelona, Spain

[*]also at: Oceanography Department, Faculty of Science, Alexandria University, Alexandria 21500, Egypt
[**]also at: Department of Environmental Sciences / Centre of Excellence in Environmental Studies, King Abdulaziz University, PO Box 80203, Jeddah, 21589, Saudi Arabia

*Correspondence to*: Marco Paglione (m.paglione@isac.cnr.it) and Manuel Dall'Osto (dallosto@icm.csic.es)

**Abstract.** Antarctica and the Southern Ocean (SO) are the most pristine areas of the globe and represent ideal places to investigate aerosol-climate interactions in an unperturbed atmosphere. In this study, we present submicrometer aerosol (PM1) source apportionment for two sample sets collected in parallel at the British Antarctic Survey stations of Signy and Halley during the austral summer 2018-2019. Water Soluble Organic Matter (WSOM) results a major aerosol component at both sites (37 and 29% of water-soluble PM1 on average, at Signy and Halley respectively). Remarkable differences between pelagic (open ocean) and sympagic (influenced by sea ice) air mass histories and related aerosol sources are found. The application of factor analysis techniques to H-NMR spectra of the samples allows the identification of five Organic Aerosol (OA) sources: two primary (POA) types, characterized by sugars, polyols and degradation products of lipids and associated to open ocean and sympagic/coastal waters respectively; two secondary (SOA) types, one enriched in methanesulphonic acid (MSA) and dimethylamine (DMA) associated to pelagic waters, the other characterized by trimethylamine (TMA) and linked to sympagic environments; and a fifth component of unclear origin possibly associated with the atmospheric ageing of primary emissions. Overall, our results strongly indicate that the emissions from sympagic and pelagic ecosystems affect the variability of submicron aerosol composition in the study area, with atmospheric circulation establishing marked latitudinal gradients only for some of the aerosol components (e.g the sympagic ones) while distributing the others (i.e., pelagic and/or aged) both in maritime as in inner Antarctic regions.

## 1 Introduction

Given their distance from major anthropogenic sources, the Southern Ocean (SO) and Antarctica are considered to be a proxy for the preindustrial atmospheric condition and processes (Cavalieri et al., 1999; Arrigo et al., 2010; Carslaw et al., 2013; Arrigo et al., 2015; Hamilton et al., 2015) which impact the climate of the entire Southern hemisphere. During winter, a layer of sea ice is formed over the SO, extending up to approximately $19 \times 10^6$ km$^2$, reducing by about 80% ($4 \times 10^6$ km$^2$) in the summer (Cavalieri et al., 1999). Climate models have large uncertainties in simulating clouds, aerosols and air-sea exchanges along with their effects on Earth's albedo in this area of the globe (Carslaw et al., 2013). One of the main reasons of this uncertainty is that the aerosol source apportionment is poorly understood. The much diverse ecosystems stretching from the SO, the sub-Antarctic marine areas to the Antarctic coastal areas and ice shelves are schematically reduced to two large natural sources governing the aerosol populations: sea spray (primary, mostly composed of sea salt) and non-sea salt sulfate (nSS-SO$_4^{2-}$; secondary). The former - with mass size distributions peaking in the supermicron range - is produced by oceanic waves breaking and bubble bursting. By contrast, the latter - secondary in type and submicron in size - is obtained by atmospheric oxidation of dimethylsulfide (DMS), a trace gas produced by marine phytoplankton. However, a recent intensification in Antarctic aerosol measurements field campaigns is revealing that aerosol chemistry in the southern high latitudes can be much more complex. For example, blowing snow over pack ice has been suggested to contribute sea salt aerosol in similar amounts to breaking waves (Legrand et al., 2017a; Giordano et al., 2018, Frey et al., 2020).

As regards of secondary (gas-to-particle formation) aerosols, the DMS-derived nSS-SO$_4^{2-}$ (Charlson et al., 1987; Vallina et al., 2007) is normally accompanied by organic sulfur species, the best known and usually more abundant of which is methanesulfonic acid (MSA) (Rankin and Wolff, 2003; Legrand et al., 2017b; Fossum et al., 2018). The role of aerosol sulfur species in regulating cloud condensation nuclei (CCN) concentrations in the marine environment is being challenged by a much larger varieties of poorly known ocean-emitted aerosol components (Quinn and Bates, 2011). Another potentially key component for new particle formation in Antarctica is iodine, which is known to form new particles via iodic acid nucleation (Saiz-Lopez et al., 2007; Baccarini et al 2021).

The scientific question about the chemical nature and source identification of the aerosols in the SO intercepts therefore the broader debate about the relative importance of secondary aerosols produced from biogenic sulfur versus primary sea-spray aerosols in regulating cloudiness in the marine environment.

Pioneering measurements of organic carbon (OC) in size-segregated aerosol (Virkkula et al., 2006) showed that MSA represented only a few percentages of the substantial amount of OC observed in the submicron fraction. Recent Antarctic measurements also suggest that the importance of organic components may have been overlooked. Saliba et al. (2021) found that the large organic fraction of particles <0.1 μm diameter may have important implications for CCN number concentrations and indirect radiative forcing over the SO. Recent measurements over the SO (43°S−70°S) and the Amundsen Sea (70°S−75°S) showed that Water Insoluble Organic Carbon (WIOC) accounted for 75% and 73% of aerosol total organic carbon in the two regions, respectively (Jung et al., 2020). In the Amundsen Sea, WIOC concentrations correlated with the biomass of a

phytoplankton species (Phaeocystis antarctica) that produces extracellular polysaccharide mucus which can be ejected by sea spray into the aerosol.

Intensified observations using advanced aerosol instrumentation onboard research ships have highlighted a certain dependence of aerosol concentrations and composition on air mass origin and atmospheric circulation patterns across latitudes. Humphries et al., (2021) identified three main aerosol source areas in the SO: northern (40-45 S), mid-latitude (45-65 S) and southern

sector (65-70 S), with different mixture of continental and anthropogenic, primary and secondary aerosols depending on the studied region. During the same period of study, Sanchez et al (2021) and Humphries et al. (2021) found a weak gradient in CCN at 0.3% supersaturation with increasing CCN concentrations to the south between 44° to 62.1° S, which may be caused by aerosol precursors from Antarctic coastal biological emissions.

At the same time, the study of the variability of aerosol sources in sub-Antarctic and Antarctic coastal areas has added

complexity to the representation of aerosol concentrations based on latitudinal changes. Indeed, emerging recent literature shows that within the polar Antarctic air masses (>60° S) aerosol populations of variable chemical composition can be observed. By analyzing simultaneous aerosol size distribution measurements at three sites, Lachlan-Cope et al (2020) showed that the dynamics of aerosol number concentrations and distributions is more complex than the simplified view of particles as composed by the sulfate–sea-spray combination, and it is likely that an array of additional chemical components and processes

drive the aerosol population. Likewise, our previous results indicate that not only the marine productivity but also the biogenic taxa and ecophysiological state of the microbiota affect the production of aerosol precursors in seawater (Dall´Osto et al., 2017; 2019). We previously showed that the microbiota of sea ice and the sea ice-influenced ocean (sympagic environment) can be a stronger source of atmospheric primary and secondary organic nitrogen (ON), specifically low molecular weight alkylamines (Dall'Osto et al., 2017; 2019) relative to open ocean areas not influenced by sea ice (pelagic ocean).  Rinaldi et

al. (2020) reported that non-methanesulfonic acid Water-Soluble Organic Matter (non-MSA WSOM) represents 6−8% and 11−22% of the aerosol $PM_1$ mass originated in open ocean and sea ice regions, respectively. This study showed that the Weddell sea areas covered by open or consolidated packed sea ice (sympagic environment) is a strong source of organic nitrogen in the aerosol. Organic nitrogen compounds should be considered when assessing secondary aerosol formation processes in Antarctica beside the known role played by sulfur aerosols (Brean et al., 2021). By means of chamber experiments

simulating primary aerosol formation *on site* in the same area around Antarctic peninsula and Weddell sea, Decesari et al. (2020) has previously reported that the process of aerosolization enriches submicron primary marine particles with lipids and sugars while depleting them of amino acids (Decesari et al., 2020). From these experiments emerged that the potential impact of the sea ice (sympagic) planktonic ecosystem on aerosol composition were overlooked in past studies, and that multiple eco-regions (sympagic environments, pelagic waters, coastal/terrestrial ecosystems) act as distinct aerosol sources around

Antarctica (Decesari et al., 2020; Rinaldi et al., 2020). In particular, Decesari et al. (2020) found at least three main bioregions sources of water-soluble organic carbon (WSOC): (1) open Southern Ocean pelagic environments dominated by primary Sea-Spray Aerosol (SSA) mainly constituted of lipids and polyols, (2) sympagic areas in the Weddell Sea, with secondary sulphur and nitrogen organic compounds and (3) terrestrial land vegetation coastal areas, traced by sucrose in the aerosol.

Here, we report atmospheric measurements during a three-months period (December 2018 - March 2019) simultaneously carried out at two Antarctic research stations (Signy and Halley). To our knowledge, this is the first study attempting aerosol characterization and source apportionment at the synoptic scale by means of parallel measurements at two distant Antarctic stations. We stress that organic water-soluble aerosol components contribute to the aerosol population and to its most hygroscopic fraction, hence we claim their overlooked climate relevant importance. Our findings highlight the heterogeneity of the Antarctic ecosystems and how this heterogeneity impacts also on the organic aerosol sources allowing also - for the first time - to report some unique insights on their space and time variability in this region of the world.

## 2 Material and methods

### 2.1 Measurement field campaigns

The measurements reported here were made in the framework of PI-ICE (Polar Interactions: Impact on the Climate and Ecology) study in the period December 2018 - March 2019 at the British Antarctic Survey's stations (BAS) of Halley and Signy. BAS Halley VI station (75°36'0" S, 26°11'0" W) is located in coastal Antarctica, on the floating Brunt Ice Shelf about 20 km from the coast of the Weddell Sea, but distant hundreds of kilometers from the open ocean (at a variable distance along the year depending on the extension of the pack ice and floating sea ice covering the Weddell in the different seasons, approximately 200km during summer). Temperatures at Halley rarely rise above 0°C and temperatures around -10°C are common on sunny summer days. Winds are predominantly from the east. Strong winds sometimes pick up the surface snow, reducing visibility to a few meters (www.bas.ac.uk/polar-operations/sites-and-facilities/facility/halley/, last visit (12/01/2024). A variety of measurements were made from the Clean Air Sector Laboratory (CASLab), which is located about 1 km south-east of the station (Jones et al., 2008). BAS Signy station at Signy Island (60°43'0" S, 45°38'0" W) is located in the South Orkney Islands (Maritime Antarctic) and is characterized by a cold oceanic climate, extremely windy, with mean annual air temperature of 3.5 C and annual precipitation ranging from 350 to 700 mm, primarily as summer rain. Summer air temperatures are generally positive (record maximum 19.8°C), although sudden falls in temperature can occur throughout the summer (-7°C has been recorded in January). Signy is also extremely windy, with prevailing westerly winds (www.bas.ac.uk/polar-operations/sites-and-facilities/facility/signy/, last visit 12/01/2024).

Two high volume samplers (MCV Barcelona Spain, equipped with Digitel $PM_1$ sampling inlet) at Signy and Halley collected ambient aerosol particles with Dp<1 μm on pre-washed (with ultrapure water) and pre-baked (at 800°C for 1h) quartz fiber filters, at a controlled flow of 500 L min$^{-1}$. Due to the necessity of collecting sufficient aerosol loading for detailed chemical analyses on the filters, the sampling time was of the order of about 50 h for each sample. A total of 8 and 14 $PM_1$ samples were collected during the field study at Halley and Signy stations, respectively. The samples were stored at about -20° C until extraction and chemical analyses.

Figure 1 shows a map of the study area. Temporal periods are reported in the Supplementary Table S1 and Figure S1 while Table S2 reports the meteorological data for the sampling periods at both sites.

## 2.2 Air mass back trajectories analysis and source regions classification

Five-days back-trajectories arriving at a height of 30 m every 6 hours were calculated using HYSPLIT - (Hybrid Single-
135 Particle Lagrangian Integrated Trajectory v4, Draxler et al., 1998; Draxler et al., 2008) and monthly Global NOAA-NCEP/NCAR pressure level reanalysis data archives. Using these, trajectory level plots were also calculated using the Openair package (Carslaw and Ropkins 2012) exploiting the Concentration Weighted Trajectory (CWT) method. The CWT approach uses the concentration measured upon a trajectory's arrival at site and the residence time of that trajectory in each grid cell it passes through to create a mean concentration for each grid cell. When plotted as a map, this shows that air masses passing
over which cells would, on average, give higher concentrations at the measurement site.

## 2.3 Aerosol offline measurements

The aerosol samples from both the sites were extracted with deionized ultrapure water (Milli-Q) in a mechanical shaker for 1 h and the water extracts were filtered on PTFE membranes (pore size: 0.45 μm) in order to remove suspended materials.
Extracts were analyzed by ion chromatography (IC) for the quantification of water-soluble inorganic ions (sodium, $Na^+$; chloride, $Cl^-$; nitrate, $NO_3^-$; sulfate, $SO_4^{2-}$; ammonium, $NH_4^+$; potassium, $K^+$; magnesium, $Mg^{2+}$; calcium, $Ca^{2+}$), organic acids (acetate, ace; formate, for; methanesulfonate, MSA; oxalate, oxa) (Sandrini et al., 2016) and low molecular weight alkyl-amines (methyl-, ethyl-, dimethyl-, diethyl- and trimethyl-amine, MA, EA, DMA, DEA and TMA, respectively) (Facchini et al., 2008a). An IonPac CS16 $3 \times 250$ mm Dionex separation column with gradient MSA elution and an IonPac AS11 $2 \times 250$
150 mm Dionex separation column with gradient KOH elution were deployed for cations and anions, respectively. The sea-salt and non-sea-salt fractions of the main aerosol components measured by IC (SS-x and nSS-x, respectively) were derived based on the global average sea-salt composition found in Seinfeld and Pandis (2016) using Na+ as the sea-salt tracer. A complete list of the species quantified by IC and used in the subsequent discussion is reported in Table S3. The data are also available at Zenodo Data public repository (doi:10.5281/zenodo.10663787).
The water-soluble organic carbon (WSOC) content was quantified using a TOC thermal combustion analyzer (Shimadzu TOC-5000A). Given MSA high relative contribution to the total organic mass, it is separated by subtracting its carbon contribution (in $\mu$gC m$^{-3}$) from total WSOC, obtaining the non-MSA WSOC. A carbon-to-mass conversion factor of 2 was used to estimate the non-MSA water-soluble organic matter (non-MSA WSOM) from non-MSA WSOC measurements, following the values suggested for marine organic aerosols by Jung al. (2020). The total WSOM was then calculated as the sum of MSA and the
non-MSA WSOM mass concentrations. Field blanks were collected at both sites and all the sample concentrations were corrected for the blanks, which resulted in negligible values.

Aliquots of the aerosol extracts were dried under vacuum and re-dissolved in deuterium oxide (D$_2$O) for organic functional group characterization by H-NMR spectroscopy (hereinafter also referred as NMR), as described in Decesari et al. (2000). The H-NMR spectra were acquired at 600MHz in a 5mm probe using a Varian Unity INOVA spectrometer, at the NMR facility of the Department of Industrial Chemistry (University of Bologna). Sodium 3-trimethylsilyl- (2,2,3,3-d4) propionate (TSP-d4) was used as an internal standard by adding 50 µL of a 0.05% TSP-d4 (by weight) in D$_2$O to the standard in the probe. To avoid the shifting of pH-sensitive signals, the extracts were buffered to pH~3 using a deuterated-formate/formic-acid (DCOO$^-$ =HCOOH) buffer prior to the analysis. The speciation of hydrogen atoms bound to carbon atoms can be provided by H-NMR spectroscopy in protic solvents. On the basis of the range of frequency shifts, the signals can be attributed to H-C containing specific functionalities (Decesari et al., 2000, 2007). The main functional groups identified include unfunctionalized alkyls (H-C), i.e. methyls (CH$_3$), methylenes (CH$_2$), and methynes (CH) groups of unsubstituted aliphatic chains (i.e., also named later "Aliphatic chains"); aliphatic protons adjacent to unsaturated/substituted groups (benzyls and acyls: H-C-C=) and/or heteroatoms (amines, sulfonates: H-C-X, with X≠O), like alkenes (allylic protons), carbonyl or imino groups (heteroallylic protons) or aromatic rings (benzylic protons) (i.e., also named later "Polysubstituted aliphatic chains"); aliphatic hydroxyl/alcoxy groups (H-C-O), typical of a variety of possible compounds, like aliphatic alcohols, polyols, saccharides, ethers, and esters (i.e., also abbreviated later as "Sug-Alc-Eth-Est"); anomeric and vynilic groups (O-CH-O), from not completely oxidized isoprene and terpenes derivatives, from products of aromatic-rings opening (e.g., maleic acid), or from sugars/anhydrosugars derivatives (glucose, sucrose, levoglucosan, glucuronic acid, etc.); and finally aromatic functionalities (Ar-H, also abbreviated later as "Arom"). Organic hydrogen concentrations directly measured by H-NMR were converted to organic carbon. Stoichiometric H/C ratios were specifically assigned to functional groups using the same rationale described in previous works (Decesari et al., 2007; Tagliavini et al., 2006): briefly, the choice of specific H/C molar ratios is based on the expected stoichiometry and structural features of the molecules that every region of the H-NMR spectra can actually represents in atmospheric aerosol samples on average. The H/C ratios used in this study are showed in Supplementary Table S4. Although the sum of NMR functional group concentrations approached total WSOC in many samples, the uncharacterized fraction was significant (on average 30 %). Possible reasons for the "unresolved carbon" are (1) the presence of carbon atoms not attached to protons, thus not-detectable to H-NMR, such as oxalates and compounds containing substituted quaternary carbon atoms or fully substituted aryls (Moretti et al., 2008), (2) the uncorrected estimations of stoichiometric H/C ratios used for the conversion of directed measured organic hydrogens into organic carbon, and (3) evaporative losses during the evaporation of the extract prior to the preparation of the NMR tube.

Organic tracers were identified in the H-NMR spectra on the basis of their characteristic patterns of resonances and chemical shifts: we used for this scope libraries of reference spectra from the literature (of standard single compounds and/or mixtures from laboratory/chamber experiments and/or from ambient field studies at near-source stations). We also validated our interpretations using extensive libraries of biogenic compounds and theoretical simulations of H-NMR spectra of atmospheric relevant molecules offered by specific elaboration tools/software such as Chenomx NMR suite (Chenomx inc., evaluation

version 9.0) and ACD/Labs (Advanced Chemistry Developments inc., version 12.01), some examples of which are reported in Supplementary (Figure S2 and S3).

Among the tracers identified, MSA and two low-molecular-weight alkyl-amines (di- and tri- methyl amines, DMA and TMA respectively) were quantified in mass concentrations. Speciation and quantification of these tracers by H-NMR were validated by comparison with the IC measurements of the same species showing excellent agreements between the two techniques (Figure S4). Other molecular tracers (such as lactic acid - Lac, betaine - Bet, choline - Cho, glycerol - Gly, glucose - Gls, sucrose - Suc, hydroxymethanesulfonic acid - HMSA) were unequivocally identified but not quantified in this study, where they are used mainly for source identification. In the present study we also refer to broadly defined chemical classes sometimes synonymously to the classes of compounds carrying specific functional groups or combinations of them, like "polyols" (i.e. compounds with NMR bands in the H-C-O region) or "saccharides" (similarly to polyols but with the concomitant presence of NMR signals in the anomeric region O-CH-O). Intense NMR bands in the H-C (unfunctionalized alkyls) region with prominent peaks characteristic of aliphatic chains (terminaly methyls at 0.9 ppm, methylenic chains at 1.2 ppm and methines or methylenes in beta position to a C=O group or an oxygen atom at 1.5 ppm) were attributed to compounds from the degradation of lipids (sometimes defined concisely as "lipids") including low-molecular weight fatty acids (LMW-FA) and mixtures of other alkanoic acids. A comprehensive list and a description of the functional groups, molecular species and categories of compounds identified in this study by H-NMR spectra analysis is reported in Table S5.

## 2.4 Factor analysis of H-NMR Spectra

The H-NMR spectra from the samples collected at both sites were analyzed by factor analysis techniques, following the method already described in previous publications (Decesari et al., 2011; Finessi et al., 2012; Paglione et al., 2014a, 2014b), to apportion major components of WSOC and attribute them to specific sources. The factor analysis was applied directly on the collection of spectra, using as input variable the spectral signals at the different chemical shifts (after several preprocessing steps described more in details in Supplementary Section S2). The factor analysis methods used in this study include two different non-negative algorithms: the "Positive matrix factorization" (PMF, Paatero et al, 1994), using the ME-2 solver (Paatero et al., 1999), and the "multivariate curve resolution" (MCR), according to the classical alternating least-square approach (Jaumot et al., 2005; Tauler 1995). The factor analysis was applied to Signy and Halley spectral datasets merged together with two main purposes: i) to increase the number of samples (14+8=22) in order to improve the statistics; ii) to find and compare relative contributions of possible common components/sources of the aerosol between the two sites.

The solutions with up to eight factors were explored. A full examination of the inputs and outcomes of the NMR factor analysis is reported in the Supplementary (Section S2, Figure S5-S12), while in section 3.3 we focus on the five-factors solution, which is the most interpretable and shows a substantial agreement between the two algorithms. Interpretation of factors and their attribution to specific sources is based on an integrated approach including: the comparison between spectral profiles and a unique library of reference spectra (recorded during laboratory studies or in the field at near-source stations, Decesari et al.,

2020); the correlation of factors contributions with available chemical tracers (i.e., sea salt and other inorganic ions, MSA and amines, Table S6); and the examination of backtrajectories and of the concentration-weighted trajectories (CWT) maps of each factor indicating their potential source areas.

Moreover, in order to specifically check the separation between primary and secondary aerosol sources, we applied the factor analysis adding to the ambient aerosol spectra also 16 H-NMR spectra of Sea-Spray Aerosol (SSA) generated in bubble bursting tank experiments by local Antarctic sea-waters and melted sea-ice during the PI-ICE project, as described by Dall'Osto al. (2022a). The results of this additional factor analysis (summarized as well in Supplementary Section S2) helped interpreting factors identified by ambient samples and attributing some of them to primary sources (POA) (Figure S11 and S12).

## 3 Results

The result section is divided in three main parts as following: in section 3.1 we discuss the bulk aerosol composition at Signy and the drivers of its variability; in section 3.2 we provide the same analysis for the data collected at Halley station and we discuss the differences between the two stations for the period of sampling overlap (42 days). Finally, in section 3.3 we discuss the WSOC source apportionment results based on the H-NMR spectra factor analysis. Samples collected at Signy are labelled as "$S_x$", whereas samples coming from Halley are labelled as "$H_x$".

### 3.1 Main aerosol chemical components at Signy

The chemical composition of the fourteen $PM_1$ aerosol samples collected at Signy station is reported in Figure 2. On average, the concentrations of the $PM_1$ aerosol water-soluble fraction are small ($1.59 \pm 1.44$ µg m$^{-3}$ average $\pm$ standard deviation, n=14) but show a noticeable variability between samples (min=0.29 µg m$^{-3}$ for S10, max=5.54 µg m$^{-3}$ for S5). The major chemical class contributing to the water-soluble $PM_1$ is sea salt (representing $45 \pm 19\%$ of the total on average of the whole sampling period) followed by WSOM ($37 \pm 19\%$, of which $6 \pm 5\%$ represented by MSA) and non-sea salt sulfate (nSS-SO4, $12 \pm 14\%$), leaving the rest to minor contributions of ammonium ($3 \pm 3\%$), nitrate ($1 \pm 1\%$) and other non-sea salt ions (i.e., nSS-K, nSS-Mg and nSS-Ca, amounting to $2 \pm 3\%$ in total).

The sampling period can be divided into two different sub-periods: the first part (samples S1-S5, corresponding to the period 10-28 Dec. 2018), is characterized by relatively high $PM_1$ concentrations ($2.75 \pm 1.96$ µg m$^{-3}$), while the second (samples S6-S14, spanning 28 Dec. 2018-15 Feb. 2019) shows lower concentrations on average ($0.95 \pm 0.37$ µg m$^{-3}$). In regards to composition, the first period is characterized by a higher contribution of sea salt (contributing $54 \pm 23\%$ to total water-soluble $PM_1$) with respect to the second period which exhibits a smaller sea salt content ($40 \pm 17\%$) and an increased fraction of non-sea salt sulfate ($19 \pm 13\%$). The contribution of WSOM shows lower variability between the two periods ($43 \pm 25\%$ and $33 \pm$

16% of water-soluble $PM_1$ in the first and second part, respectively, of which MSA represents $1 \pm 1\%$ and $8 \pm 5\%$, respectively).

Nevertheless, the inter-samples variability of the WSOM concentrations is large ($0.49 \pm 0.35$ μg m$^{-3}$ on average, min=0.18 μg m$^{-3}$ for S10, max=1.25 μg m$^{-3}$ for S5). The main difference in WSOM at Signy between the two subperiods stands in the functional group composition, as characterized by H-NMR analyses (Figure 3). Specifically, the first sub-period (S1-S5) is enriched in alcoxy groups (H-C-O) ($43 \pm 4\%$ relative to $23 \pm 8\%$ of total WSOC) and unsubstituted aliphatic chains (H-C) ($30 \pm 5\%$ relative to $16 \pm 5\%$) with respect to the second subperiod (S6-S14). These H-NMR features have been previously

associated - by comparing the analysis of tank-generated sea-spray particles - to sugars, polyols (e.g., glucose, sucrose, glycerol, etc.) and fatty acids from lipids degradation of primary biogenic origin (sea and sea-ice microbiota) (Facchini et al., 2008b; Decesari et al., 2011; Decesari et al, 2020; Liu et al., 2018; Dall'osto et al., 2022a; Dall'Osto et al., 2022b). By contrast, the second period is enriched by MSA ($24 \pm 13\%$ relative to $3 \pm 3\%$ in the first period in term of WSOC as reconstructed by H-NMR) and alkyl-amines ($15 \pm 8\%$ relative to $2 \pm 1\%$) which are considered mostly secondary in nature (Dall'Osto et al.,

2019).

The meteorological conditions are not statistically different between the two periods (Table S2). The analysis of the air masses origin (Figure S13) on the contrary reveals different back-trajectories between the first five samples (S1-S5) and the last nine ones (S6-S14) suggesting that the observed changes in the chemical composition are linked to a different origin of the air masses reaching the sampling site, as we further discuss in the next sections.

### 3.2 Main aerosol chemical components at Halley and comparison with Signy

The chemical composition of the eight $PM_1$ aerosol samples collected at the Halley station (reported in Figure 4) shows remarkable differences with respect to Signy. Overall, water-soluble $PM_1$ mass concentrations are substantially lower ($0.79 \pm 0.56$ μg m$^{-3}$ average $\pm$ standard deviation, n=8) and shows a smaller variability between samples (min=0.25 μg m$^{-3}$ for H4,

max=2.02 μg m$^{-3}$ for H7). Most noticeably, a much lower contribution of sea salt to $PM_1$ is measured in these samples, representing on average only $6 \pm 7\%$ in striking contrast with Signy. Only one sample (H5) shows a relatively high influence of sea salt (23% of total $PM_1$). In contrast to Signy, the $PM_1$ chemical composition at Halley is constantly dominated by non-sea salt sulfate and WSOM, representing on average $53 \pm 17\%$ and $29 \pm 14\%$ (of which $8 \pm 6\%$ represented by MSA), respectively. In the samples collected at the end of January, nSS-sulfate represents a major component of $PM_1$ at both Halley

and Signy but its concentrations at the former site are greater, peaking above 1 μg m$^{-3}$. In summary, based on the analysis of the submicron aerosol bulk composition, a first sharp difference between the two sites can be underlined, with Signy being much more impacted by primary aerosol (sea-spray) while Halley by secondary ones (nSS-SO4).

The organic composition expressed in terms of H-NMR functional groups and molecular tracers shows a limited variability between samples at Halley, with a dominant contribution of alcoxyl groups (H-C-O), MSA and alkyls (H-C) representing on

average $33 \pm 7\%$, $28 \pm 14\%$ and $18 \pm 6\%$, respectively (Figure 5). The significant contribution from MSA is in line with the

high nSS-sulfate shared in the water-soluble PM$_1$ mass, while the high contributions from H-C-O and H-C groups indicate a contribution from primary aerosol which does not show up from the inorganic composition data alone.

The contribution of MSA at Halley, representing the $28 \pm 14\%$ of total WSOC on average, is along the whole sampling period a significant aerosol mass contribution regardless Halley is quite distant from open ocean regions. Alkyl-amines instead represent only a minor portion of WSOC in Halley samples ($3 \pm 2\%$ on average) contrary to the parallel samples in Signy ($14 \pm 8\%$). A direct comparison of the average water-soluble PM$_1$ and WSOC concentrations and composition at Signy and Halley, based on the samples collected in parallel during 42 days of campaign, is reported in Figure 6 (only the samples of the second sub-period of Signy are considered here, hence excluding S5). Although the total concentration of water-soluble PM$_1$ in the parallel samples is comparable between the two sites, the average values reported in Figure 6a highlight once more the much higher contribution of sea salt at Signy (contributing $0.40 \pm 0.29$ µg m$^{-3}$, representing on average $42 \pm 18\%$ of total WS PM$_1$ during the overlapping period) with respect to Halley, which on the contrary was dominated by nSS-SO$_4$ ($53 \pm 17\%$ of total WS PM$_1$). WSOM represented in any case the second major component of PM$_1$, with similar proportions between the two sites but also with remarkable differences in the functional groups' distribution and tracers' concentrations (Figure 6b). A lower contribution of alcoxyl groups and higher concentrations of alkyl-amines is observed at Signy with respect to Halley. All these differences strongly suggest different sources and origin areas for the aerosol collected at the two sites, which is also confirmed by the analysis of the air masses showed in Figure S14. The analysis of the back-trajectories highlights a much stronger influence of marine air masses in Signy, in agreement with the substantially higher contribution of sea salt in PM1 and the reduced secondary organic nitrogen compounds (alkyl-amines) in Halley. At the same time, however, the MSA concentrations are similar between the two sites in spite of the remoteness of Halley from the sea, and the nSS-SO$_4$ concentrations – which are known to be impacted by the DMS sources in the ocean – are greater in Halley than in Signy. The MSA/nSS-SO$_4$ ratios are $0.55 \pm 0.23$ at Signy and $0.24 \pm 0.05$ at Halley (on average of the overlapping period). The Halley values are in line with previous measurements at the same site by Legrand et al. (1998). At Signy the MSA/nSS-SO$_4$ is consistent with the latitudinal trend evidenced by Bates et al. (1992) for the southern hemisphere.

Finally, the organic composition in Halley rich of H-C-O and H-C groups points to a primary organic contribution which again conflicts with the prevalently continental back-trajectories reaching the Halley station. Some clues for resolving such discrepancies are provided by the organic factor analysis discussed in the next section.

### 3.3 WSOA Source Apportionment: POA & SOA types and their contributions

The large inter-sample variability in the H-NMR spectra characterizing the two sample sets of Signy and Halley makes factor analysis a potentially powerful tool for source identification and source apportionment in spite of the small numerosity of samples. A full examination of the outcomes of NMR factor analysis is reported in the Supplementary (Section S.2, Figure S5-S12). Here we report a description of the 5-factors solution which was identified as the most robust and informative one (Figure 7) based on the best separation of interpretable spectral features and on the best agreement between the two algorithms

applied with respect to both spectral profiles and contributions. The associated concentration-weighted trajectories (CWT) maps for each factor, indicating their potential source areas, are reported in Figure 8. Our factor analysis was able to identify two Primary Organic Aerosol (POA), two Secondary OA (SOA) and another factor prevalently found at Halley and of unknown origin. Specifically, the WSOM factors are:

- Factor 1 - "marine POA pelagic (lipids, polyols and saccharides)", characterizing most of the samples both at Signy and Halley and found in comparable concentrations and relative contributions among the parallel samples at the two sites. Interestingly this factor is characterized by an NMR spectral profile dominated by polyols (glycerol and possibly threitol) and saccharides, (found in some samples as glucose and sucrose) together with aliphatic compounds bearing alkyl chains, such as in low-molecular-weight fatty acids (LMW-FAs) . All these features are typical of primarily emitted submicron sea-spray particles generated by bubble bursting experiments of biologically-productive sea-waters, as already documented in previous studies conducted in similar Southern Ocean areas (Decesari et al., 2020; Dall'Osto et al., 2022a) but also in North Atlantic Ocean (Facchini et al., 2008b). This POA component was present in almost all the samples (especially at Signy) and were prevalently associated with air masses coming from open ocean regions, including large sectors of the SO north-western of Signy and eastern of Halley (see the CWT maps in Figure 8). The CWT maps for Factor 1 in Halley, although showing large overpasses over continental Antarctica clearly show a maximum when being influenced by a northern origin in the SO. In summary, this POA seems to be a common component of the sea-spray OA associated to open ocean areas across a wide range of longitudes and can be transported for thousands of kilometers.

- Factor 2 - "marine POA (Lac)", representing a significant portion (up to 70%) of WSOC in specific samples especially at Signy (i.e., S3-S5). It shows a mixture of LMW-FAs and polyols, similar to Factor 1, but with an important contribution from lactic acid (Lac, peaks at 1.35 and 4.21 ppm in the H-NMR spectra). Lactic acid – a major product of sugars fermentation common to many microorganisms (Miyazaki et al., 2014) – was already identified in sea-water and sea-spray aerosol samples of the region and considered of primary biogenic origin (Decesari et al., 2020; Dall'Osto et al., 2022a). In particular, these features are characteristics of specific sea-spray aerosol samples coming from sea-water bubble bursting experiments conducted in coastal areas around Adelaide Island, Davis Coast and Livingston Island (SW3, SW4, SW8, and especially SW11, SW15 in Figure S8, S11 and S12). This factor is found only in the first subperiod of Signy (samples S1-S5) and the associated CWT maps are similar but somewhat showing higher concentrations closer to the site with respect to Factor 1. For this reason, we consider it as a second marine POA factor influenced by aerosol sources in coastal/sympagic areas around the Antarctic Peninsula.

- Factor 3 - "marine SOA pelagic (MSA+DMA)", dominated by methanesulfonic acid and dimethyl amine (identified by the H-NMR singlets at 2.80 ppm and 2.71 ppm, respectively). The predominance of these compounds indicates a

marine biogenic secondary formation for this factor. The contributions time series of Factor 3 shows a good correlation with the concentrations of MSA at both sites ($R^2$=0.88, n=22, p<0.005) and nSS-SO$_4$ in particular at Halley ($R^2$=0.58, n=8, p<0.1). The CWT maps associated with Factor 3 (Figure 8) at Signy shows the predominance of air masses coming from the open Southern Ocean and spending most of the time on pelagic waters (Figure S15). At Halley instead, this component shows maxima in air-masses originating in open-ocean areas at North-East but travelling above the Planetary Boundary Layer or PBL (Figure S16) and possibly reaching the station through free-tropospheric circulation spending time over the Antarctic continent. In summary, this component can be considered as a background/regional marine SOA source associated with emissions in pelagic waters of the SO.

- Factor 4 - "marine SOA sympagic (TMA+MSA)", characterized by high loadings of trimethyl amine, even higher than MSA. This component was very characteristic of Signy (absent in Halley) and especially of the second sampling period at that site (S6-S14). The corresponding CWT maps clearly assign Factor 4 to a source footprint stretching over the sympagic waters of the Weddell Sea. This observation agrees with our previous findings in the same area pointing to sympagic Weddell sea region as a source of biogenic organic nitrogen and in particular amines in ambient aerosols (Dall'Osto et al., 2017; Dall'Osto et al., 2019; Decesari et al., 2020; Brean et al., 2021).

- Factor 5 - A final factor was found to characterize the organic composition in Halley. It accounts for a substantial fraction of the H-C-O and H-C groups at Halley, corresponding as sum to the contribution from the pelagic POA (Factor 1) but with distinct groups of H-NMR resonances. In particular, a complex pattern of H-NMR signals is found at chemical shift between 4 - 4.5 ppm (in the range of the H-C-O groups) (Figure S17). These signals have never been observed before in ambient aerosol samples and are largely missing in the Signy samples. They can be tentatively attributed to acidic sugars (e.g., uronic acids) or organic sulfate (sulfate-esters), as better discussed in Supplementary (Figure S10 and corresponding text). Considering the high abundance of nSS-SO4 and the likely corresponding acidic nature of the aerosol in Halley, a hypothesis for the formation of these compounds can be the esterification of common polyols (such as glycerol) to organic sulfates. However, this hypothesis is just speculative at this stage and possibly needs confirmation from additional analysis/data. As already mentioned, alkoxyl groups are usually considered primarily emitted (confirmed also by the presence of degraded/oxidized lipids signals at 0.9, 1.3, and 1.6 ppm in the alkyls region of the Factor 5 spectral profile) but the hypothesized substitution of hydroxyls with sulfonate groups can be considered of secondary nature (ageing of primary alcohols/sugars). Moreover, the factor profile shows contemporary some other secondary features, such as MSA and DMA signals which makes this factor of difficult identification. For this reason, we consider it as a mixture of primary and secondary OA ("POA-SOA mix") characterizing Halley site. A clearer interpretation of the nature of this organic fraction could not be achieved at the moment. The CWT maps for Factor 5 do not elucidate any specific source areas. It is speculated that it could be a

mixture of primary and secondary components and partially potentially coming from continental/terrestrial environments (Kyrö et al. 2013).

Figure 7 reports the average contributions of the five factors at the two sites. It is worth noting how the WSOA factor analysis confirmed the importance of primary sources (POA) in the first sampling period at Signy already evidenced by the bulk chemical analyses: the sum of POA factors in fact represents 89% of the total WSOA on average of the first 5 samples (S1-S5). Much more relevant are the contributions of secondary components (SOA) in the second sampling period at Signy (71% of total WSOA on average of samples S6-S14) and at Halley (35% of WSOA on average). The dominant component at Halley remained in any case the so called "POA-SOA (mix)" factor accounting for the 44% of WSOA at the site. Excluding Factor 5, the trend of the contribution of POA vs SOA is clear: the SOA fraction of OA increases while the austral summer progresses, as a possible consequence of the increasing emissions of reactive vapors from the ocean together with enhanced photochemistry. Nevertheless, one clear episode of high POA concentration is observed in the middle of January at both stations, indicating that synoptic circulation can augment the transport of POA from the Southern Ocean over the entire Weddell Sea region and into the Antarctic continent across a 2,000 km -wide area even in the middle of the Austral summer.

## 4 Discussion

The variability of the aerosol populations in the polar southern latitudes is affected by strong latitudinal changes in both aerosol sources and atmospheric circulation. Humphries et al. (2021) evidenced, in the area of East-Antarctica, latitudinal gradients in atmospheric aerosol loading and composition which were put in relation with the position of the atmospheric polar front. Our results extend such observations to the West Antarctica region of Peninsula and Weddell sea, while adding new insights on the nature of aerosol sources and the drivers of aerosol chemistry.

The air mass back trajectories travelling to Signy and Halley, reported in Figure S14, as well as CWT maps in Figure 8, clearly show how the two sites are mostly influenced by different air mass origin and history. In particular, Signy (60°S, in the maritime Antarctica) is impacted by two types of air masses: one (being prominent in the first part of the campaign, samples S1-S5) associated with the Westerlies, and spending most of the time on pelagic waters of the SO; the other (influencing the second period, samples S6-S14) recirculating over the Weddell sea and spending more time over sympagic waters and sea-ice marginal zones (Figure S13-S15). By contrast, Halley (75°S, over the ice shelf) is mainly affected by an anticyclonic flow (from E or SE) over the Antarctic continent (60% of the air mass recirculated over the Antarctic continent), involving air masses having travelled over consolidated packed ice with a minor influence from the pelagic environments in the SO (Figure S14 and S15). Whilst the Signy samples were representative of air masses that had previously travelled almost entirely within the PBL, different conditions were observed at Halley, where only 59±24% travelled within the PBL (Figure S16a,b). Since the specific eco-regions (sympagic, pelagic) supply aerosol populations with distinct physical and chemical characteristics

(Dall'Osto et al., 2017; Dall'Osto et al., 2019; Decesari et al., 2020; Rinaldi et al., 2020; Brean et al., 2021), we show here that the latitudinal change between 60°S and 75°S in the prevalent atmospheric circulation tends to maintain a segregation between the specific aerosol populations produced in the different environments. This explains why primary sea salt particles are found in much greater amounts in Signy and aged nSS-SO4 particles affect Halley to a greater extent. However, the results of the organic factor analysis highlight a more complicated picture. Summarizing our findings on the inorganic and organic characterization at the two stations, we can distinguish three atmospheric regimes at least:

(a) During the subperiod in Signy (Dec), $PM_1$ particles transported by the Westerlies and coming from pelagic waters of the open Southern Ocean (organic Factor 1) with some possibly affected by marine ecosystems in the coastal Antarctica (organic Factor 2) increased the water-soluble PM1 concentrations above 1 (up to 5) µg m$^{-3}$ and were mostly contributed by primary constituents (sea salt and marine POA components);

(b) During the second subperiod in Signy (Jan – Feb), the aerosol originated prevalently from sympagic areas of the Weddell sea region and was characterized by lower concentrations (1 µg m$^{-3}$ or lower) and by a higher contribution of secondary components (nSS-SO$_4$ and SOA, especially enriched in biogenic organic nitrogen and in particular TMA);

(c) In Halley (Jan), water-soluble PM1 occurred at very low concentrations (only occasionally reaching 1 µg m$^{-3}$) and were dominated again by secondary components, especially nSS-SO4, MSA and DMA, but also by OA classes specifically found at this site and of unclear origin.

The relatively high concentrations of nSS-SO4 in the sinking anticyclonic air masses arriving to Halley point to processes of atmospheric ageing for this site. This may explain the lower MSA/nssSO4 ratio at Halley, assuming partial MSA oxidation to nssSO4 during transport. However, nSS-SO4 is associated to MSA and DMA, i.e. organic components overlapping to the "pelagic SOA" of Signy (and classified accordingly in Factor 3) meaning that the products of oceanic emissions find their way into the free troposphere and lead to aerosol formation in the Antarctic continental atmosphere. This study highlights again the importance of amines and organic nitrogen in SOA formation in southern polar areas, as already evidenced by our previous studies in the same area (Dall'Osto et al., 2017 and 2019). We identify here two factors of organic nitrogen. One - rich of TMA - is associated with sources in the sympagic waters of the Weddell Sea (Factor 4). A second one (characterized by DMA and MSA) is associated with air masses from pelagic open SO waters (Factor 3). Factor 3, in particular, appears to be a background component of the Antarctic atmosphere in the middle of the austral summer (Jan) across latitudes (as it affects both Signy and Halley) and it is linked to long-range transport and to marine emissions in a wide source area (Figure 8). Factor 4, instead, tracing SOA formation from emissions in the Weddell Sea, mainly affects the maritime western Antarctica, not Halley in spite of the proximity of the site to the source regions. This can be explained by the fact that Halley does not receive direct flows from the Weddell Sea itself but rather from coastal Antarctic areas in eastern longitudes where sea ice is much less developed in summer (Figure S14-S15).

The occurrence of specific organic factors at one site and not in the other (Factors 2 and 4 uniquely in Signy, and Factor 5 uniquely in Halley) is in line with the idea that atmospheric circulation maintains chemical gradients in the Antarctic aerosol

populations. Nevertheless, we show that the secondary components associated with Factor 3 (MSA and DMA) can cross such gradients and be distributed at different latitudes. Figure 9 shows the average concentrations of the WSOA components identified by NMR factor analysis on average at Signy (both as average of the whole sampling period and of the parallel

samples) and Halley stations. Factors 1 and 3 share a common time trend of contribution to WSOA at the two sites, suggesting that they represent background aerosol spread around a wide area of thousands of kilometers. Such common background components are of secondary and, more surprisingly, primary marine origin. The factor "marine SOA pelagic (MSA+DMA)" has very similar concentrations at Signy and Halley ($3.21\pm2.15$ and $2.45\pm2.21$ nmolH m$^{-3}$ respectively, on average for the parallel sampling period) (Figure 9). Most noticeably, the same is true for the "marine POA pelagic" factor ($1.60\pm1.20$ and

$1.84\pm3.11$ nmolH m$^{-3}$ at Signy and Halley, respectively). At Halley, "marine SOA pelagic" is associated to air masses travelling above the PBL (Figure S16b) and its concentration correlates with those of nSS-SO$_4$, supporting the hypothesis of a long-range transport associated to a free-tropospheric flow over the Antarctic dome. By contrast, the "marine POA pelagic" factor did not correlate with nSS-SO$_4$ and is associated to strong winds and to air masses coming directly from NE, especially during the episode of mid-January. Therefore, the "marine POA pelagic" factor is not associated to the free-tropospheric circulation and

rather reached the station through transport in the PBL from the Southern Ocean sectors located in the NE to the site (Figure 8). Contrary to Halley, the Signy site is sometimes dominated by other POA types which looked to have more local origin and an influence from the sympagic and/or coastal environments, such as the "marine POA (lac)" and the "marine SOA sympagic (TMA+MSA)" factors.

When considering the primary aerosol components, their concentrations normalized by sea salt can be further informative on

their origin/importance. It can be observed that the background "marine POA pelagic" enrichment in sea-spray particles is constant through all the dataset at Signy (0.05 in the first period vs 0.04 in the second). This supports the hypothesis of a constant contribution of the background POA pelagic component in sea-spray to which other primary components can be added from local biologically productive waters, as shown by samples S1-S5. In such samples, indeed, the background "marine POA pelagic" fraction contributes less than 40% to the total POA, mostly represented by the other primary component "marine

POA (lac)" (Figure 7). In these samples the total POA/sea salt ratio reached up to 0.14 (averagely). Conversely, the background POA-pelagic was more enriched with respect to sea salt at Halley (average value of 0.19), where it is also the dominant POA component. This may depend on different sea-spray production conditions (e.g., reduced salinity close to the shelf or in polynias) or to some ageing process during transport to the station which removes preferentially the more soluble inorganic fraction of sea-spray. These hypothetical mechanisms remain just speculative at this stage and needs further investigations,

but in any case, it is worth noting that marine POA may influence a much wider geographical area than the simple sea salt concentrations would suggest.

# 5 Conclusions

The Antarctic ecosystems are characterized by a substantial spatial heterogeneity across marine (pelagic and sympagic) and terrestrial biomes, with productivity and biodiversity patchiness superimposed to strong environmental gradients (Convey et al., 2014). This study represents the first chemical characterization and source apportionment of organic aerosol conducted in parallel at the two British Antarctic stations of Signy and Halley, representing two different Antarctic environments separated by 2000 km exposed to different but partly overlapping biogenic sources. In contrast to the paradigm of reducing the aerosol composition in background Antarctic regions to sea spray (primary, mostly composed of sea salt) and non sea salt sulfate ($nSS$-$SO_4^{2-}$; secondary), we find that Water Soluble Organic Matter (WSOM) is the second most abundant submicron aerosol component in this area of the world, accounting for a substantial fraction of the total water-soluble $PM_1$ mass, both at Signy (37%, min-max 16-71%, after sea salt 45%, min-max 9-80%) and Halley (29%, min-max 7-44%, after non sea salt sulfate 53%, min-max 29-83%). Our results starkly highlight how the heterogeneity of the Antarctic ecosystems impact also on the organic aerosol sources allowing - for the first time - to report some unique insights on their space and time variability in that region of the world.

In particular, significant differences are found between pelagic (characterized by higher $PM_1$ concentrations and more primary components) and sympagic (dominated by secondary components and in particular amines) periods at Signy and at Halley. The sympagic area of the Weddell Sea appear to be a strong source of Organic Nitrogen compounds in the maritime Antarctica (Signy) and in particular of low-molecular weight amines, confirming the results of previous studies in the same area (Dall'Osto et al., 2017; Dall'Osto et al., 2019; Decesari et al., 2020; Rinaldi et al., 2020; Brean et al, 2021). The amines speciation among samples from the different sites and over a longer period highlight that TMA is dominant over Weddell-sympagic waters (specifically characterizing Signy during the second part of the measurement period) while DMA is spread on a larger scale, reaching Halley (regional background footprint, similarly to MSA and non-sea salt sulfate).

Enclosed between the Antarctic continent and the pack sea-ice of the Weddell sea, Halley station shows a distinct chemical composition, much depleted of sea salt and enriched in $nSS$-$SO_4$ with respect to Signy, likely due to long-range transport and ageing in the free troposphere. The "chemical segregation" of Halley prevents inputs of certain OA types found in Signy including SOA produced by emissions in the Weddell Sea, but also allows specific aerosol organic compounds (possibly associated with organic sulfate) to develop in Halley and not in Signy.

A part from such differences between the two environments, our study highlights the existence of background biogenic marine sources, which influence the aerosol composition on a larger scale (regional or even supra-regional): among these in particular there is a secondary marine component of pelagic origin ("Marine SOA pelagic (MSA+DMA)"), but also noteworthy a marine primary source ("marine POA pelagic (lipids, polyols and saccharides")), which seems to travel for long distances across latitudes. In particular, the sinking FT air masses arriving in Halley are shown to transport SOA originating from marine emissions of DMS and DMA in distant oceanic regions, and that the prevalent atmospheric flow in Halley is occasionally interrupted by the direct transport of POA (from emissions in the SO) in the PBL.

In conclusion, our study contributes to highlight the striking complexity of the aerosol sources in a natural/pristine environment such as the Antarctic ecosystems. Ongoing climate change is predictable to change the Antarctic environment (Rintoul et al., 2018), which in turn will feedback to biosphere and cryosphere exchanges with the atmosphere, changing the atmospheric concentrations of aerosols and cloud condensation nuclei (CCN) with a yet unknown further climate feedback. Future interdisciplinary studies using emerging chemical and statistical analytical techniques are required to tease out processes across spatial gradients of key environmental factors.

**Data availability**

Data discussed in the manuscript are available at https://zenodo.org/records/10663787 (Zeonodo data repository, doi: 10.5281/zenodo.10663786; Paglione et al., 2024).

**Competing interests**

The authors declare that they have no conflict of interest.

**Author contributions**

M.D., A.J and M.Ri. designed the research; M.D., A.J., and M.Ri. organized the field campaigns; M.D., and A.J. collected the aerosol samples; M.P., F.M., and M.Ru. performed the chemical analyses; M.P. performed the factor analysis of the H-NMR spectra; S.D., A.M., E.T, contributed to the H-NMR spectra interpretation and to the factor analysis discussion and correction; D.C.S.B, and K.M elaborated back-trajectories and maps. M.P., M.D., M.Ri., and S.D. wrote the paper. RMH, TL-C and all the authors contributed the scientific discussion and paper revision.

**Acknowledgments**

The study was supported by the Spanish Ministry of Economy through project PI-ICE (CTM2017–89117-R) and POLAR-CHANGE (PID2019-110288RB-I00). The National Centre for Atmospheric Science NCAS Birmingham group is funded by the UK Natural Environment Research Council. Financial support was provided also by the European Commission: H2020 Research and innovation program, project FORCeS (grant no. 821205). This work acknowledges also the 'Severo Ochoa Centre of Excellence' accreditation (CEX2019-000928-S)

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

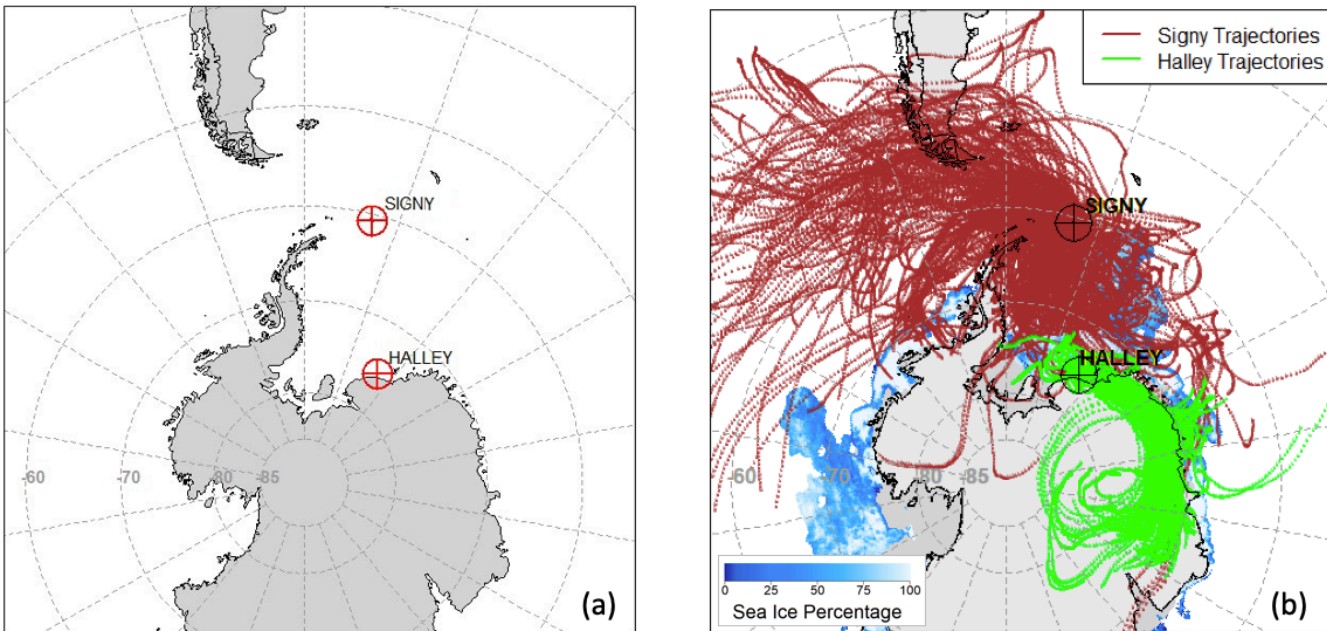

**Figure 1. (a) Maps of the study area with BAS Signy and Halley stations and (b) air mass back trajectories for the all the sampling periods at both stations. Bluish areas represent the sea-ice cover averaged over the studied period. Grey areas out of the continental boundaries represent the average shelf-ice cover.**

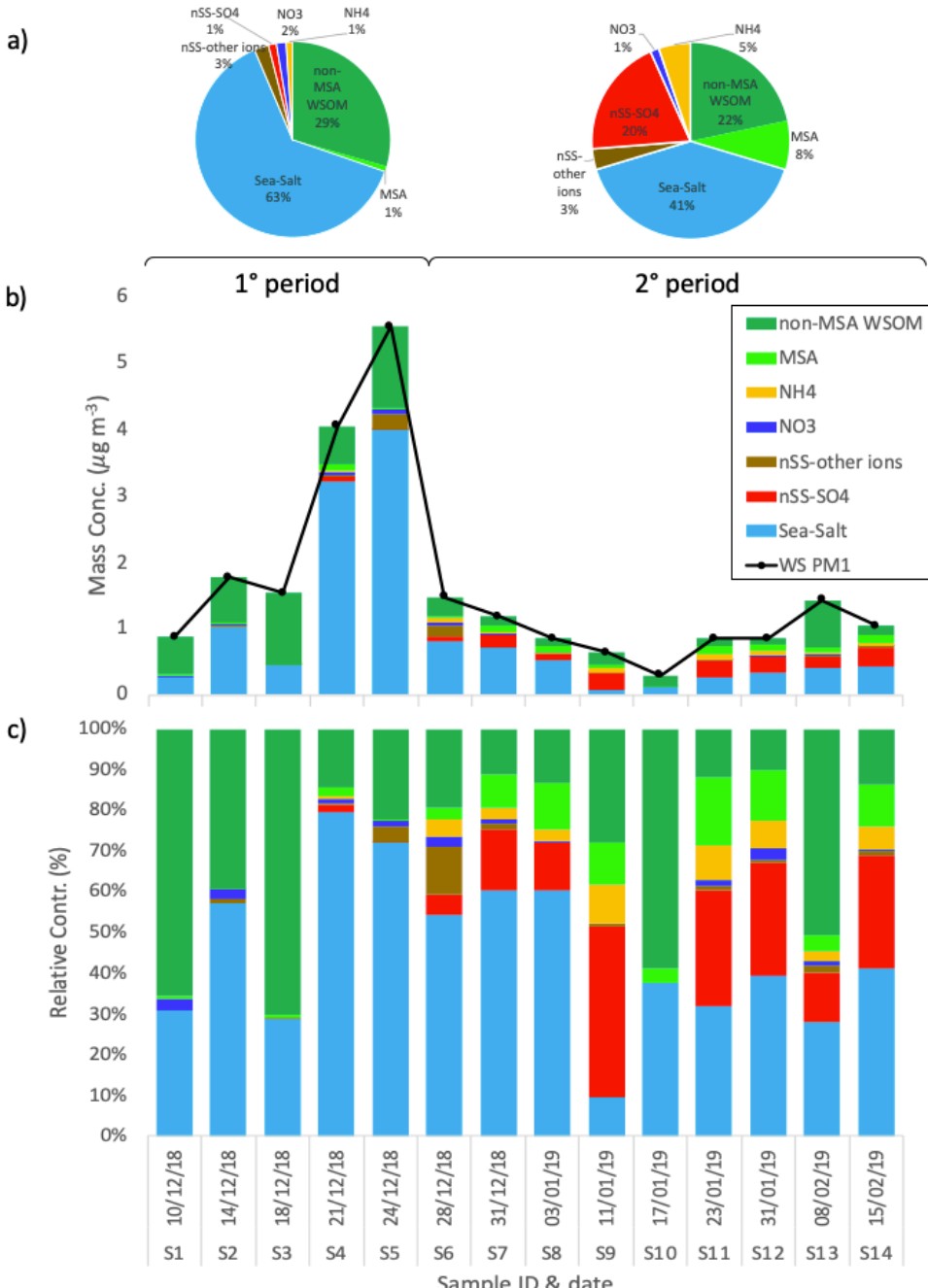

**Figure 2. Water-soluble PM$_1$ loadings and chemical composition at Signy during the whole period. Pie charts in panel a) report the average relative contributions for two different periods of the campaign: 1° period (samples S1-S5) and 2° period (samples S6-S14). Panel b) and panel c) show respectively the mass concentrations and the relative contributions of the different chemical species measured in each sample.**

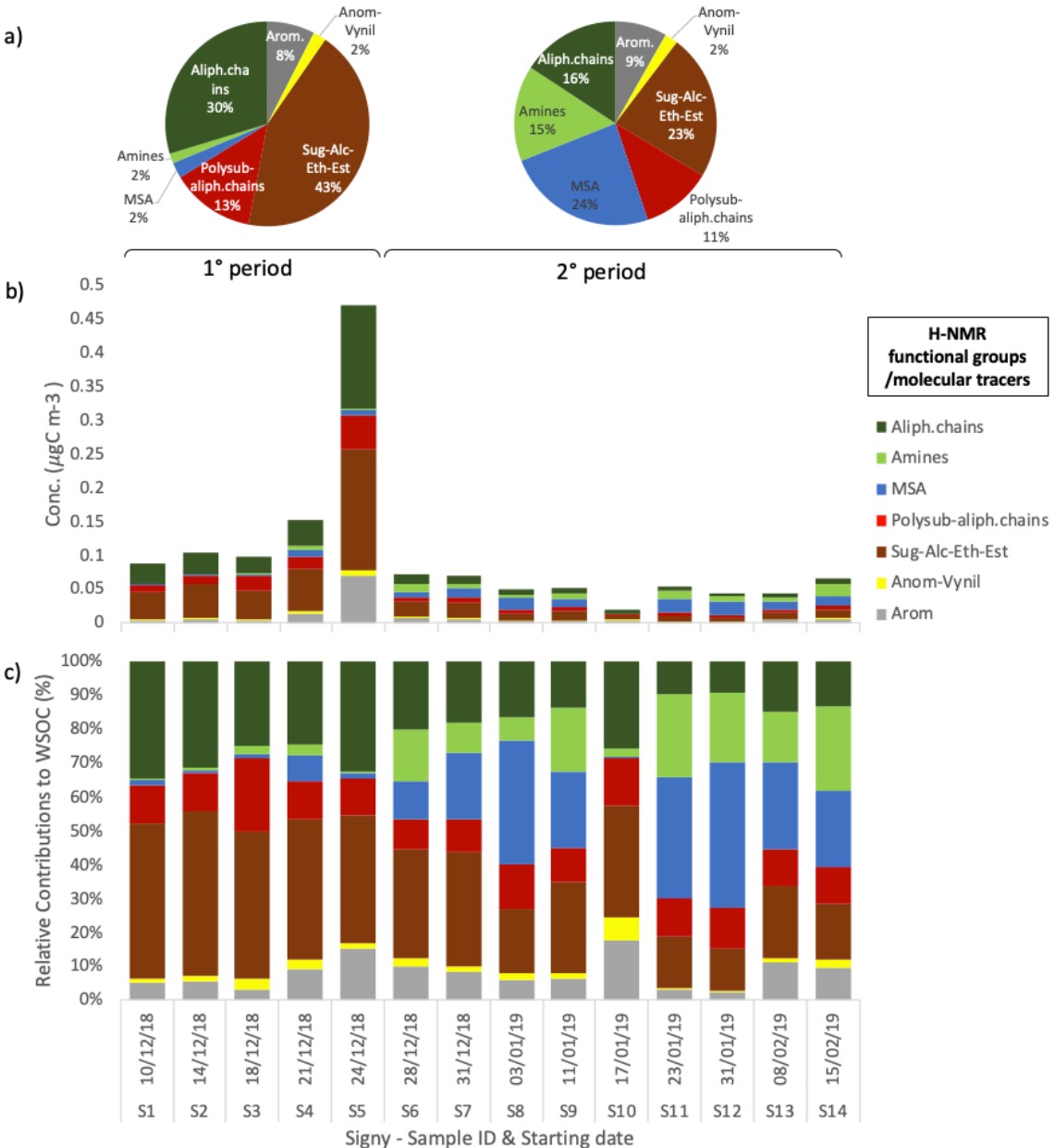

**Figure 3. Water-soluble OC concentrations and composition in term of H-NMR functional groups at Signy. (a) average relative contributions of two different periods of the campaign: 1° period (samples S1-S5) and 2° period (samples S6-S14). Panel b) and panel c) show respectively the mass concentrations and the relative contributions of the different functional groups identified and quantified by H-NMR in each sample (expressed in μgC m⁻³).**

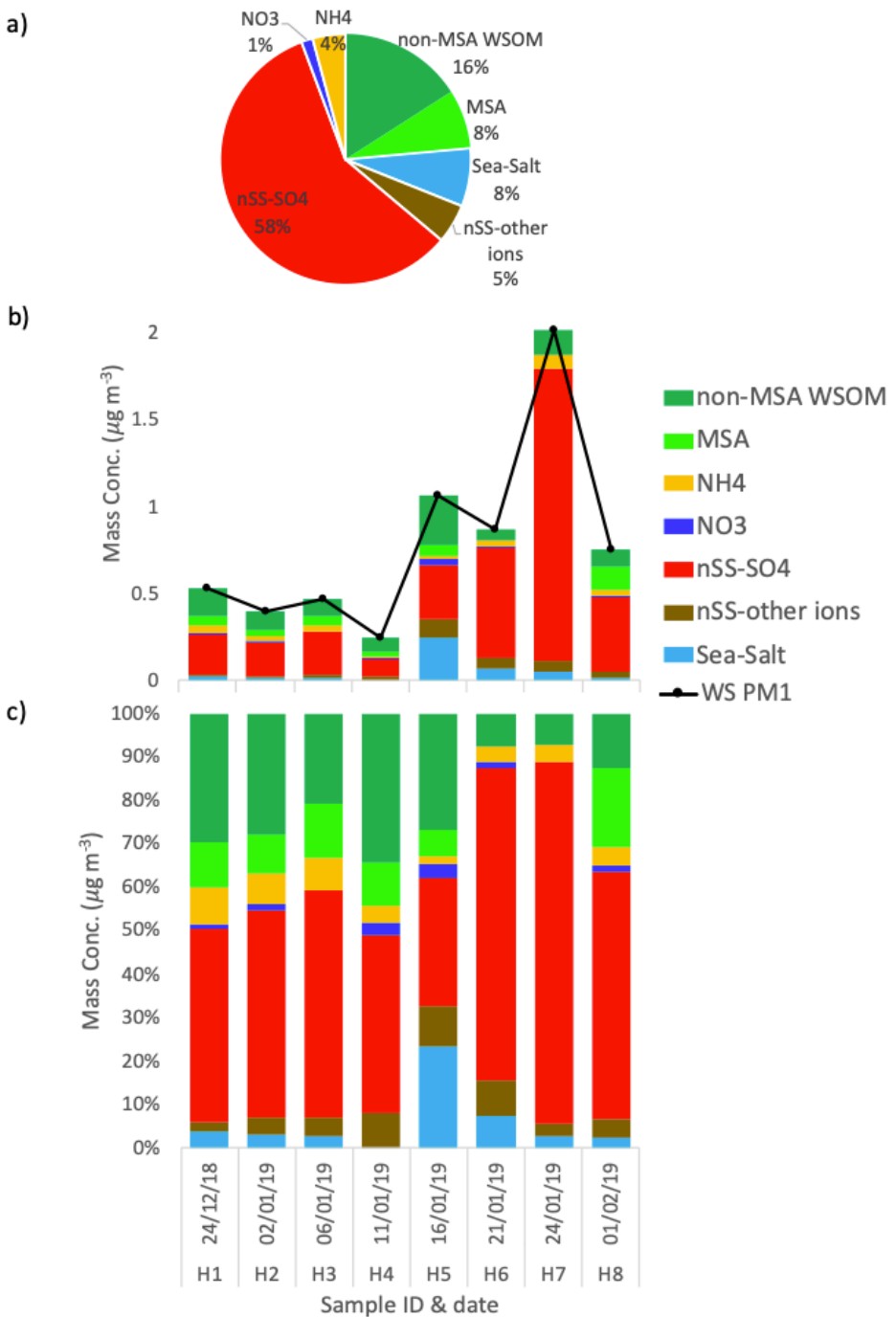

Figure 4. Water-soluble PM$_1$ loadings and chemical composition at Halley. Pie charts in panel a) represent the average relative contributions of the whole sampling period, while panel b) and c) respectively the mass concentrations and the relative contributions of the different chemical species measured in each sample.

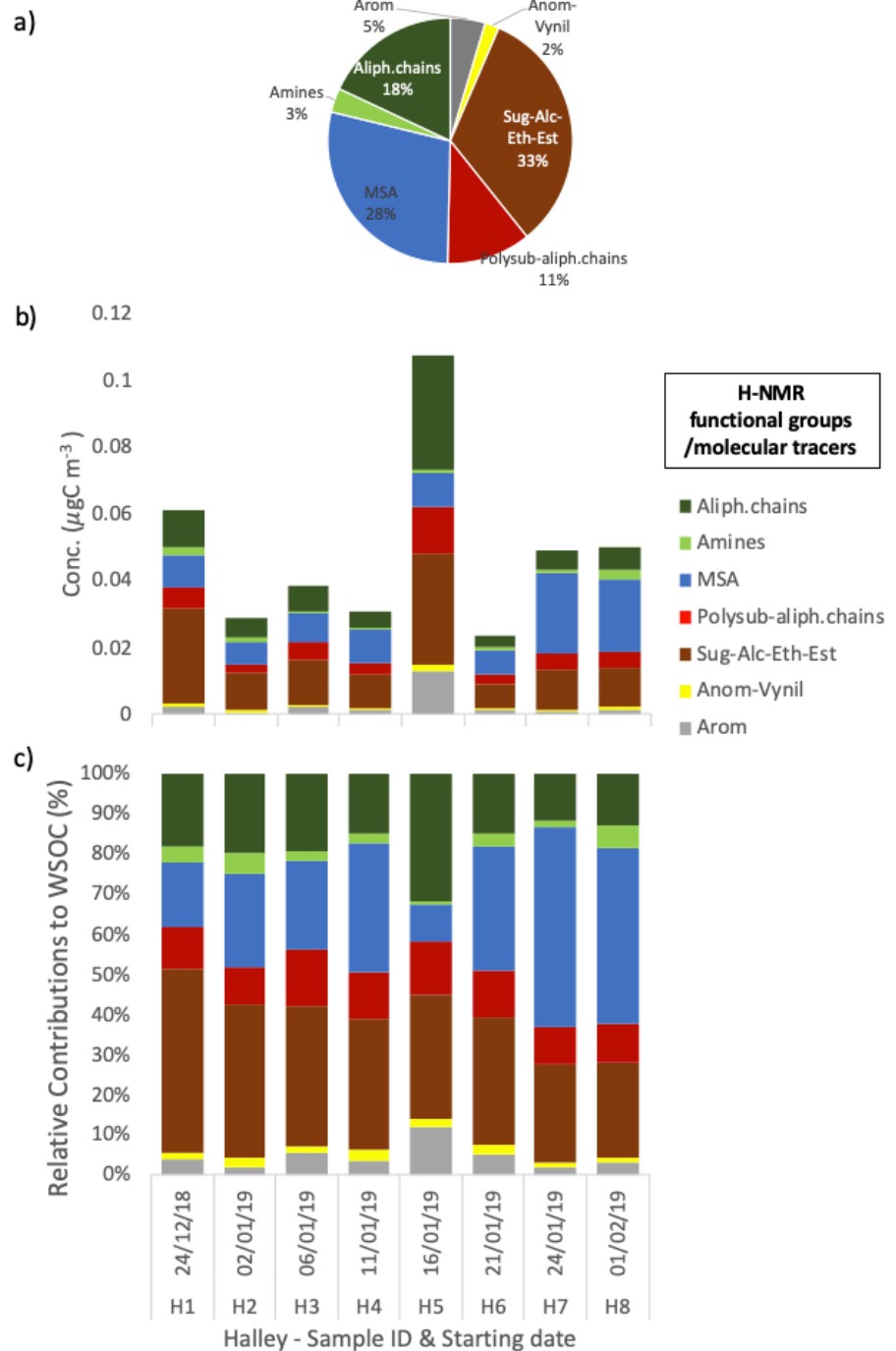

**Figure 5.** Water-soluble OC concentrations and composition in term of H-NMR functional groups at Halley. (a) average relative contributions; (b) and c) show respectively the mass concentrations and the relative contributions of the different functional groups identified and quantified by H-NMR in each sample (expressed in µgC m⁻³).

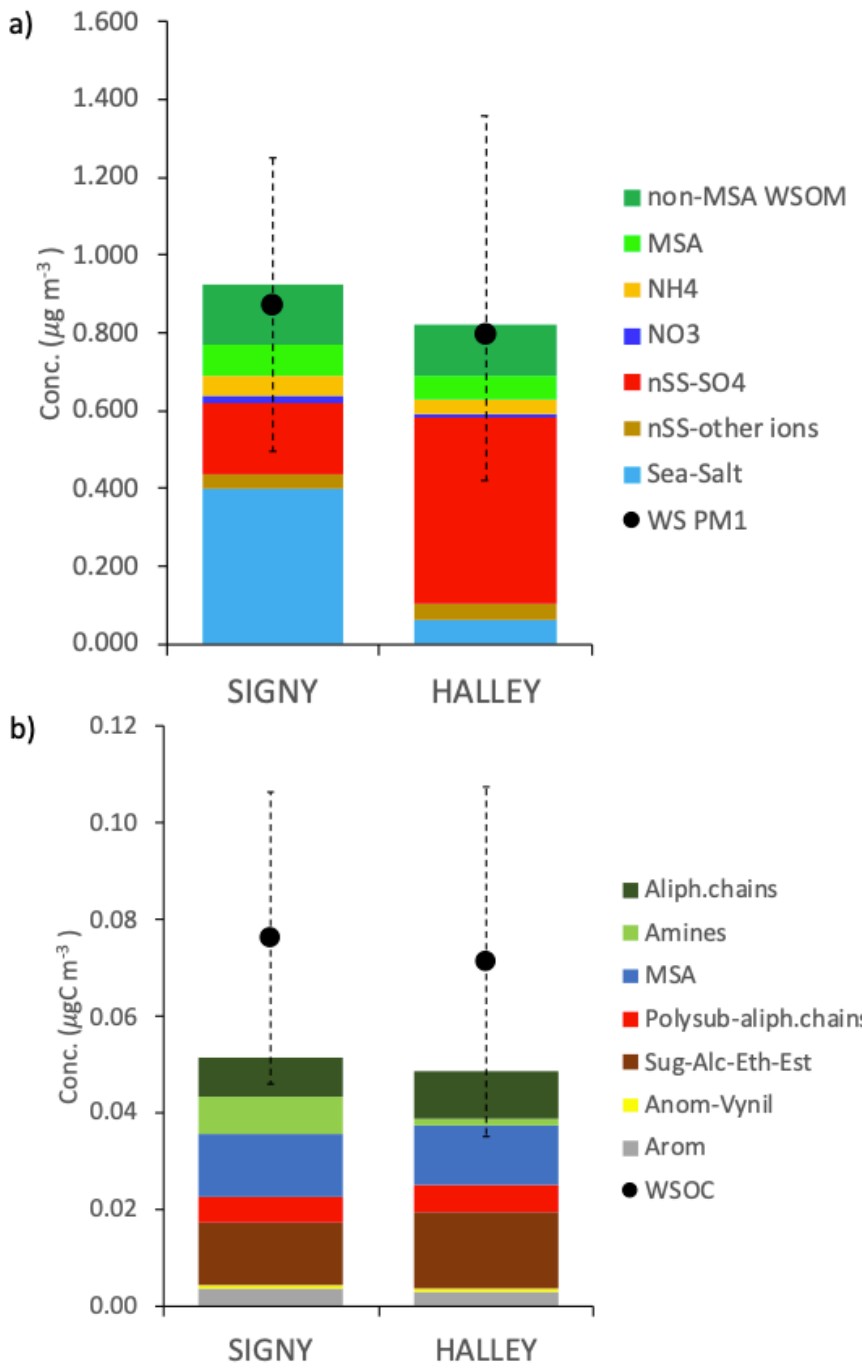

**Figure 6. Average concentrations at Signy and Halley for the overlap period (S6-S12 and H1-H8). (a) average concentrations of water-soluble PM₁ and its main components; (b) Average concentrations of WSOC measured by TOC and H-NMR functional groups concentrations**

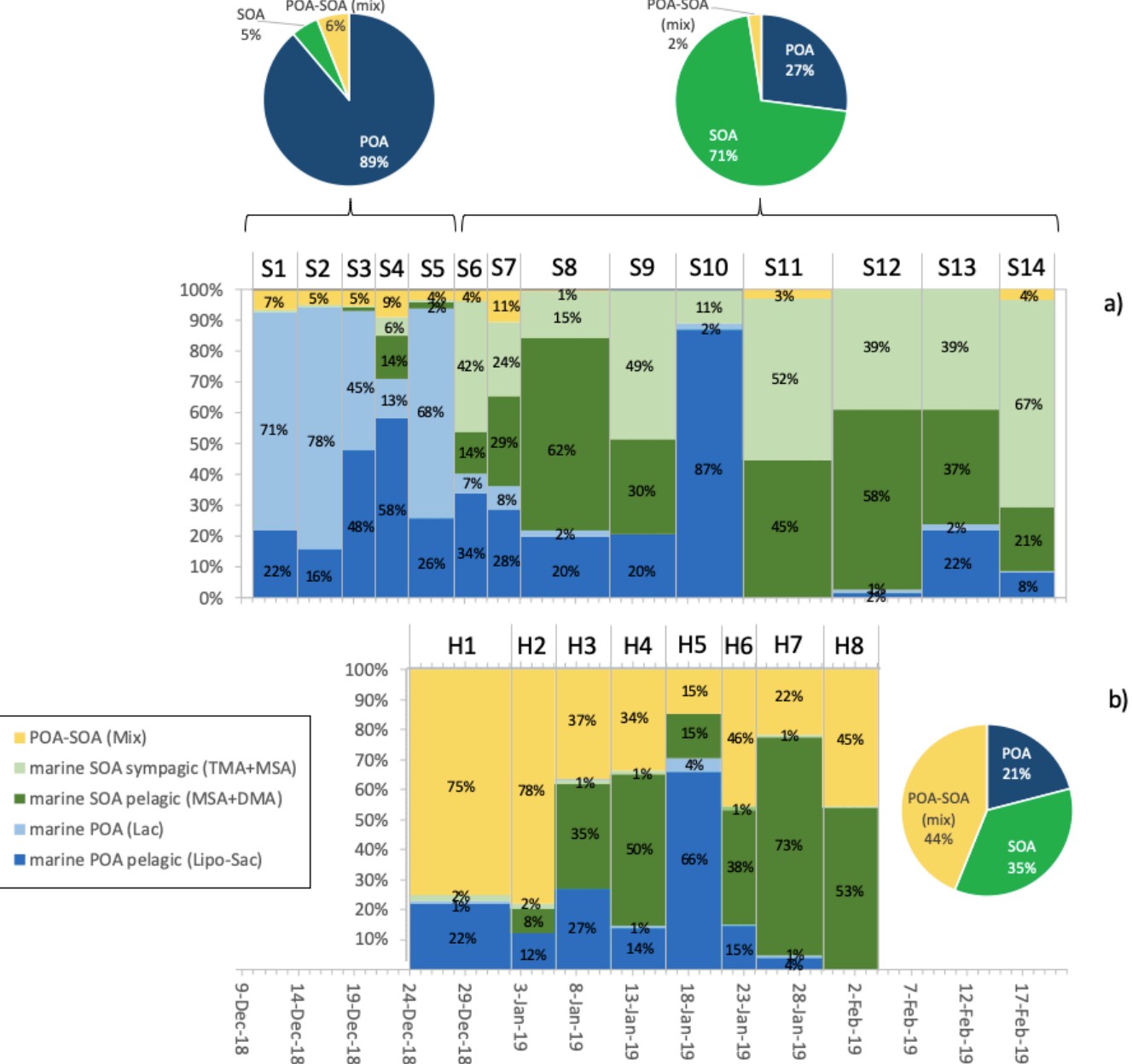

**Figure 7. Relative contributions of the WSOA factors identified by H-NMR factor analysis at Signy (panel a) and Halley (panel b). Histograms show the relative contributions (%) of each factor in each sample: blueish colours refer to POAs component, while greenish colours to SOA components; in yellow the POA-SOA (mix) factor. Pie charts report the average values of the sum of factors classified as POA (dark-blue), SOA (green) and POA-SOA mix (yellow) at the two sites and in different periods.**

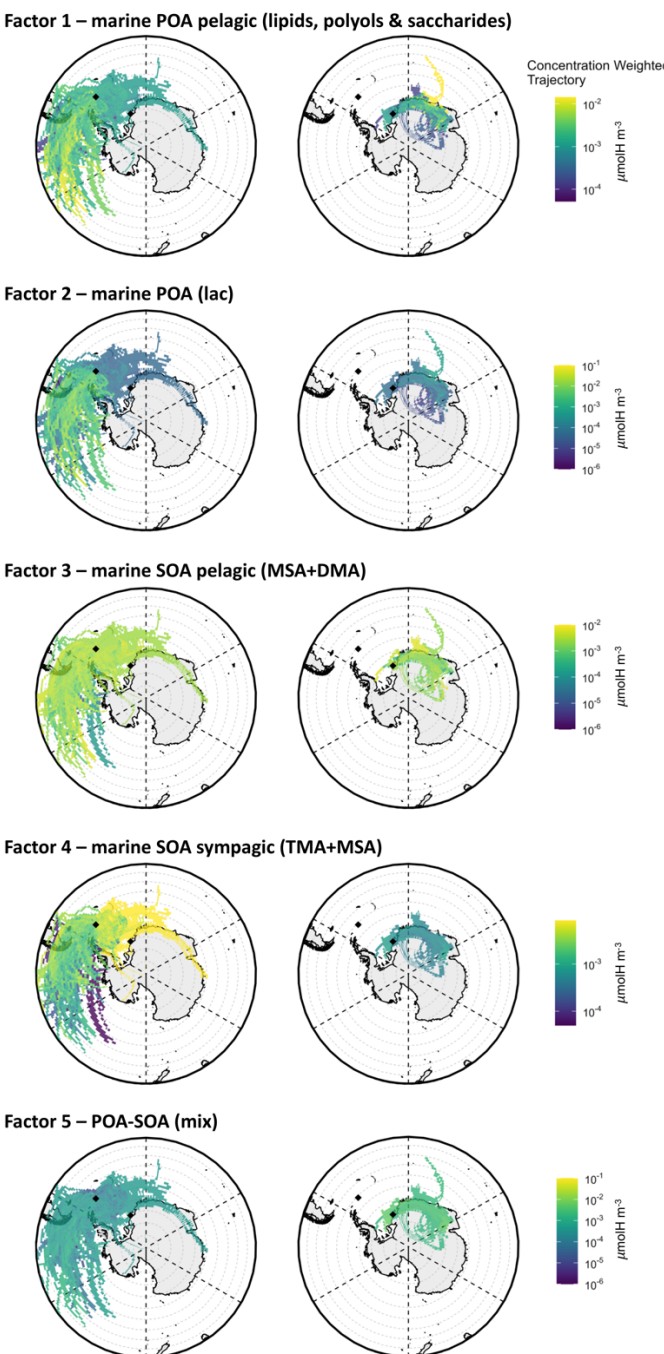

**Figure 8. CWT maps of the WSOA components identified by the factor analysis of NMR-spectra at Signy (left-side) and Halley (right-side). Colour scale shows which air masses along the backtrajectories give, on average, higher concentrations (expressed in $\mu molH\ m^{-3}$) at the measurement site: lighter colours (toward yellow) and darker colours (toward blue) indicate respectively higher or lower concentrations of the component associated to the air masses coming from a specific area.**

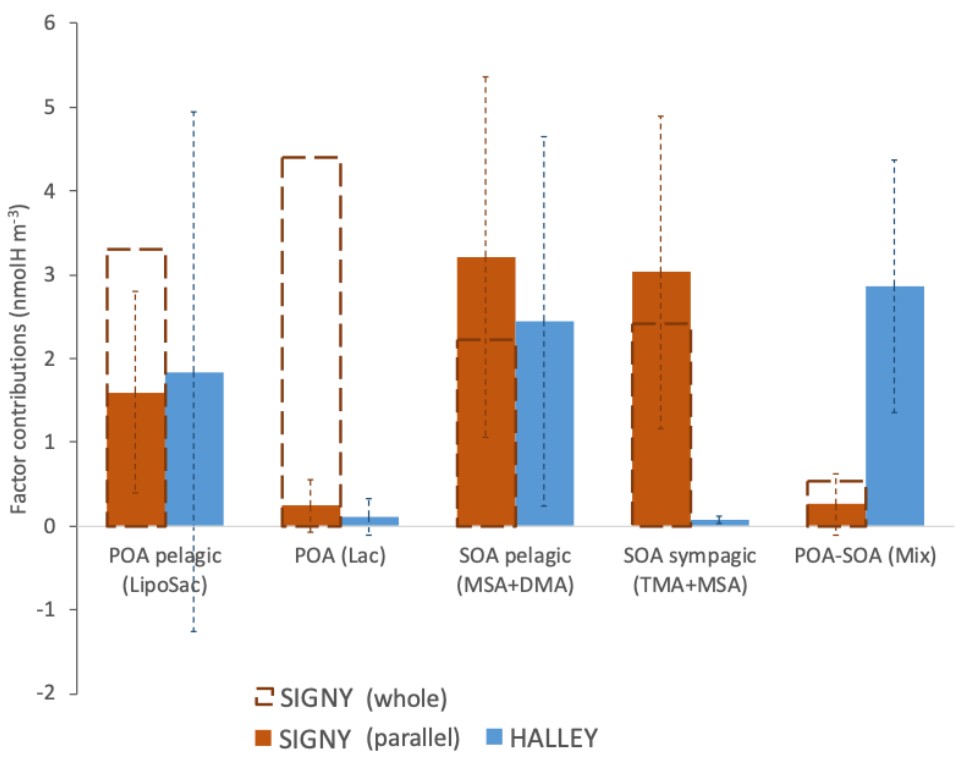

**Figure 9. Average contributions of WSOA components identified in the overlapping period at the two Antarctic Stations.**

830