# Peer review of "Simultaneous organic aerosol source apportionment at two Antarctic sites reveals large-scale and eco-region specific components"

_EGUsphere, 2023_

## Author Comment (AC1)

We thank the Reviewer #1 for the helpful and constructive comments, which have led to significant improvements in the manuscript. We have carefully revised the text. Our point-by-point replies are given below (blue), following the referees' comments (black). Changes to the manuscript are marked with green. In our replies, the line numbers (when reported) refer to the revised manuscript.

**Reviewer 1**:
Paglione et al. describe the chemical composition of aerosol particles (PM1) collected at two Antarctic stations (Signy and Halley) during a field campaign from December 2018-March 2019. For the chemical analysis of the organic and inorganic compounds they applied several types of offline methods, such as H-NMR and ion chromatography. They observed differences in the chemical composition based on the dominance of either open ocean or sea ice in the air mass history. By applying non-negative matrix factorization, they identified five sources for organic aerosol relevant to this polar marine region.

Ambient aerosol particles contain a plethora of organic molecules, where a majority still eludes chemical characterization and quantification using classical instrumental approaches. The use of H-NMR appears to me unconventional, but promising with the large potential to enhance the knowledge about chemical compounds in aerosol particles. Even though, I am aware that the colleagues from Bologna are skilled and experienced in the use of NMR, I am very impressed, partially sceptic, with which confidence the authors attribute signals to individual compounds in the 'forest' of NMR signals in ambient samples, where potential interferences from other chemical compounds in the NMR spectra are very likely.

I believe that the scientific community could largely benefit from the findings of this study. A revised manuscript should be eventually published in ACP. However, I noticed large gaps in confirmability and thoroughness, which makes it hard to readers without profound knowledge in NMR analysis to understand (and get convinced of) the train of thoughts of the authors from their observations to their scientific conclusions. To enhance accessibility, I encourage the authors to conduct a meticulous revision, addressing these gaps and providing clarity in their reasoning from observations to scientific conclusions.

Response: We thank the Reviewer for the supportive comments and the constructive criticisms.
We recognize the difficulty and the potential skepticism about extracting molecular-level chemical information from the complexity of the NMR spectra of environmental matrices, such as those of atmospheric organic aerosol. Nonetheless, the employ of NMR spectroscopy for characterization of atmospheric organic matrices is not new and actually now exists a quite large literature targeting several specific applications, encompassing functional group analysis, identification of chemical classes and molecular identification, which was recently summarized in a review paper just issued (Decesari et al., 2024). The attributions and quantifications proposed in the present manuscript are indeed based on methodologies developed in 25 years of NMR research.
Anyway, we thank the Reviewer for highlighting the lack of clarity in the presentation of the NMR methodology used in this study. We realized that the nomenclature adopted in the Supplementary material had not been edited carefully and this certainly has caused confusion in attentive readers like the Reviewer. Based on his/her comments and suggestions we have revised the manuscript trying to enhance its accessibility, as detailed below.

**General concerns:**
The presentation of chemical compounds in this study lacks clarity, resulting in a somewhat chaotic manuscript. Figures 2, 4, and 6 highlight various substances such as WSOM, ammonium, nitrate, 'nSS other ions', nSS sulfate, and sea salt. Figures 3 and 5 introduce amines, MSA, and organic functional groups, including unsubstituted aliphatic groups, polysubstituted aliphatic groups, and anomeric and vinylic groups. Figure S3 identifies lactic acid, low-molecular fatty acids, glucose,

sucrose, 'generic polysaccharides' (What is that?), acidic sugars, neutral sugars, glycerol, MSA, TMA, and DMA. Figure S10 includes HMSA and 'acidic sugars (e.g., uronic acids) or sulfonate-esters'. Additionally, the written text discusses 'threitol' (Line 265), 'oligomers (such as betaine)' (SI, Line 60), 'glycolipids and phospholipids' (SI, Line 61), 'lipopolysaccharides' (SI, Line 72), and 'lipids and polyols' (SI, Line 74). To enhance the clarity and organization of this information, I suggest incorporating two comprehensive tables into the manuscript. The first table should summarize substances identified through chromatographic analyses, while the second should focus on substances identified through H-NMR analysis. The proposed H-NMR tables could include columns for the 'name of the substance/functional group', 'chemical shifts used for identification', 'chemical shifts used for quantification', 'examples for molecules', and 'references'. This structured approach would greatly improve the accessibility and overall understanding of the diverse chemical compounds identified in this study.

Response: we thank the Reviewer for highlighting this flaw of clarity. Certainly, the nomenclature of chemical classes presented in the Supplementary material was not carefully edited resulting in confused or inconsistent terminology. Following his/her suggestion we carefully revised text, tables and figures trying to give more systematic and homogeneous definitions of chemical substances and categories measured and discussed. We have also agreed to add in the Supplementary two new Tables reporting a list and a description of the main chemical species/categories measured by chromatographic and spectroscopic analyses used in the manuscript (new Table S3 and S4, also reported below). We agree that the reader needs support in understanding the reasoning behind the use of NMR spectroscopy for the identification of individual compounds and chemical classes in organic aerosol samples. We have added more information in Table S4 and S5.

Table S3. Ion Chromatography measured species list

| ions name | ions ID | category | sea-salt components* | non sea-salt components** |
|---|---|---|---|---|
| acetate | ace | organicanions | | |
| formate | for | organic anions | | |
| methan-sulfonate | MSA | organic anions | | |
| chloride | Cl | inorganic anions | SS_Cl | nSS_Cl |
| nitrate | NO3 | inorganic anions | | |
| sulfate | SO4 | inorganic anions | SS_SO4 | nSS_SO4 |
| oxalate | oxa | organic anions | | |
| sodium | Na | inorganic cations | SS_Na | nSS_Na |
| ammonium | NH4 | inorganic cations | | |
| methyl-amine | MA | organic cations | | |
| ethyl-amine | EA | organic cations | | |
| potassium | K | inorganic cations | SS_K | nSS_K |
| di-methyl-amine | DMA | organic cations | | |
| di-ethyl-amine | DEA | organic cations | | |
| tri-methyl-amine | TMA | organic cations | | |
| magnesium | Mg | inorganic cations | SS_Mg | nSS_Mg |
| calcium | Ca | inorganic cations | SS_Ca | nSS_Ca |

*the main ions constituting sea-salt are calculated and grouped based on the global average sea-salt composition found in Seinfel&Pandis, 2016. Briefly, Na concentrations are considered to come entirely from sea-salt. Then, starting from Na concentrations the other sea-salt components are calculated by the relative contribution to the total based on the average global composition of sea-salt (Seinfeld and Pandis, 2016). Finally, the total sea-salt is the sum of the different sea-salt components.
**non sea-salt components are calculated for each species subtracting the sea-salt part from the total concentrations

**Table S5**. H-NMR identified/measured functional groups/chemical species/categories. *Functional groups are in *italic*. **Categories including some of the other species specifically identified are in underlined italic

| name of the species/ functional group*/ category of compounds** | ID of the species/ functional group | chemical shifts used for identification & quantification | examples for molecules | possible origin/source | references |
|---|---|---|---|---|---|
| *aromatic protons* | Ar-H | band 6.5-8.5 ppm | phenols, nitro-phenols [...] | biomass burning, [...] | Decesari et al., 2001; Tagliavini 2006; Decesari et al., 2007; Chalbot and Kavouras, 2014 |
| *anomeric and/or vinyl protons* | O-CH-O | band 6-6.5 ppm | vinylic protons of not completely oxidized isoprene and terpenes derivatives, of products of aromatic-rings opening (e.g., maleic acid), or anomeric protons of sugars derivatives (glucose, sucrose, levoglucosan, glucuronic acid, etc.) | biogenic marine mostly primary | Decesari et al., 2001; Claeys et al. 2004; Schkolnik & Rudich, 2005; Tagliavini 2006; Decesari et al., 2007; Chalbot and Kavouras, 2014 |
| *hydroxyl/alkoxy groups* | H-C-O | band 3.2-4.5 ppm | aliphatic alcohols, polyhols, saccharides, ethers, and esters | biogenic marine primary | Chalbot and Kavouras, 2014 |
| *benzyls and acyls/ amines, sulfonates* | H-C-C= / H-C-X (X≠O) | band 1.8-3.2 ppm | protons bound to aliphatic carbon atoms adjacent to unsaturated groups like alkenes (allylic protons), carbonyl or imino groups (heteroallylic protons) or aromatic rings (benzylic protons) | biogenic/anthropogenic mostly secondary | Decesari et al., 2001; Graham et al., 2002; Decesari et al., 2007; Chalbot and Kavouras, 2014 |
| *unfunctionalized alkylic protons* | H-C | band 0.5-1.8 ppm | methyls (CH3), methylenes (CH2), and methynes (CH) groups of several possible molecules: fatty acids chains, alkylic portion of biogenic terpenes, etc. | biogenic/anthropogenic primary/secondary | Decesari et al., 2001; Graham et al., 2002; Decesari et al., 2007; Chalbot and Kavouras, 2014 |
| hydroxymethansulfopnic acid | HMSA | singlet at 4.39 ppm | | anthropogenic secondary | Suzuki et al., 2001; Gilardoni et al., 2016; Brege et al 2018 |
| methane-sufonate | MSA | singlet at 2.80 ppm | | biogenic marine secondary | Suzuki et al., 2001; Facchini et al., 2008a; Decesari et al., 2020 |
| di-methylamine | DMA | singlet at 2.72 ppm | | biogenic marine secondary | Suzuki et al., 2001; Facchini et al., 2008a |
| tri-methylamine | TMA | singlet at 2.89 ppm | | biogenic marine secondary | Suzuki et al., 2001; Facchini et al., 2008a |
| *N-osmolytes* | | singlets between 3.1 and 3.3 | betaine, choline and other structurally similar N-containing compounds not unequivocally identified (e.g., phosphocholine) | biogenic marine primary | Cleveland et al., 2012; Chalbot et al., 2013; Decesari et al., 2020; Dall'Osto et al., 2022b |
| betaine | Bet | singlet at 3.25 ppm (not quantified here but possibly quantifiable) | | biogenic marine primary | Cleveland et al., 2012; Chalbot et al., 2013; Decesari et al., 2020; Dall'Osto et al., 2022b |
| choline | Cho | singlet at 3.18 ppm (not quantified here but possibly quantifiable) | | biogenic marine primary | Cleveland et al., 2012; Chalbot et al., 2013; Decesari et al., 2020; Dall'Osto et al., 2022b |
| *saccharides* | Sac | used synonymously for compounds carrying H-C-O groups in unresolved mixtures but when also anomeric protons (O-CH-O) are present | glucose, sucrose and other sugars structurally similar not unequivocally identified | biogenic marine primary | Graham et al., 2002; Facchini et al., 2008b; Decesari et al., 2011; Decesari et al, 2020; Liu et al., 2018; Dall'osto et al., 2022a |
| glucose | Gls | anomeric doublet at 5.22 ppm & specific structures between 3.5 and 4.2 ppm (not quantified but possibly quantifiable @5.22 ppm) | | biogenic marine primary | Decesari et al., 2020; Dall'Osto et al., 2022b |
| sucrose | Suc | anomeric doublet at 5.40 ppm & specific structures between 3.5 and 4.2 ppm (not quantified but possibly quantifiable @5.40 ppm) | | biogenic marine primary | Decesari et al., 2020; Dall'Osto et al., 2022b |
| *polyols* | | unresolved mixture not quantified (including glycerol and D-threitol) | glycerol, threitol, erytritol and structurally similar molecules not unequivocally identified | | |
| glycerol | Gly | specific structures at 3.55, 3.66 & 3.77 ppm (not quantified but possibly quantifiable @ 3.55 ppm) | | biogenic marine primary | Decesari et al., 2020; Dall'Osto et al., 2022b |
| D-threitol | | specific structures between 3.6 - 3.7 ppm (not quantified) | | biogenic marine primary | suggested in this study (to be confirmed) |
| *acidic-sugars / sulfonate esters* | | band 4-4.3 ppm (not quantified) | uronic acids, sulfonate-derivatives of polyols | biogenic marine primary/secondary | suggested in this study (to be confirmed) |
| *neutralsugars (saccharides) and polyols* | | band 3.5-3.9 ppm (not quantified) | glucose, sucrose and other sugars structurally similar not unequivocally identified | biogenic marine primary | Graham et al., 2002; Facchini et al., 2008b; Decesari et al., 2011; Decesari et al, 2020; Liu et al., 2018; Dall'osto et al., 2022a |
| *low-molecular weight fatty acids or "lipids"* | LMW-FA | unresolved complex resonances at 0.9, 1.3, and 1.6 ppm in the H-C spectral region | fatty acids (free or bound) from degraded/oxidized lipids (e.g. caproate, caprylate, suberate, sebacate, etc.) and similar compounds owning a chemical structures of alkanoic acids. | biogenic marine primary | Graham et al., 2002; Facchini et al., 2008b; Decesari et al., 2011; Decesari et al, 2020; Liu et al., 2018 |
| lactic acid | Lac | doublet 1.37-1.36 ppm & quadruplet at 4.23 ppm (not quantified but possibly quantifiable @1.37-1.36 ppm) | | biogenic marine primary | Suzuki et al., 2001; Decesari et al., 2020; Dall'Osto et al., 2022a |

To reply point-by-point to the raised unclear definitions/explanations:
-regarding the species showed in Figure 2, 4 and 6, following also the suggestion of Referee#2, we changed the text about ion chromatography and TOC measurements in section 2.3 as follow:

[revised manuscript text omitted]

In general, in modern high-field NMR spectroscopy, even 1 D techniques can be accurate for molecular-level identification in complex matrices when well-resolved individual resonances match the spectra of standard compounds. The employ of a pH buffer can overcome the chemical shift variability of the resonances of weak acids and bases. When the overlap with background signals is such that the resolution of specific resonance is imperfect, then the match with the spectra in the libraries can be obtained only tentatively, and we have stated this clearly in the text (as in the case of threitol). In other parts of the discussion, the spectra of standard compounds were used only to characterize spectral bands in the samples in certain ranges of chemical shifts without attempting any attribution to individual compounds or families of homologous compounds: for instance, the spectra of uronic acids and sulfonate-esters were used to formulate hypotheses about the origin of the system of peaks between 4.0 and 4.3 ppm in the spectra of the Halley samples, and we concluded that the attribution of such resonances in the spectra of the samples to acidic sugars (broadly defined) and/or sulfonate esters of sugars is possible but remains uncertain. Finally, the identification of broad spectral bands to functionalities and chemical classes is largely based on the comparison with the available spectra for mixtures of POA and SOA obtained in laboratory conditions (reaction chambers, bubble-bursting tanks etc.) (Decesari et al., 2024). Such identification is not free from interference and is context-specific. For instance, the attribution of alkylic groups in the samples showing broad peaks at 0.9 ppm (terminal methyls), 1.2 ppm (methylenic chains) and 1.5 ppm (methynes in branched

structures, or methylenes in beta position to C=O groups) to the linear aliphatic chains of mixtures of alkanoic acids like low-molecular weight fatty acids and other products originating from the hydrolysis or degradation of lipids is based on the characteristic spectral features obtained for bubble-bursting aerosols (Facchini et al. 2008; Decesari et al 2020). In principle, such attribution cannot be considered free of interference, as linear aliphatic structures can originate also from pollution sources, but it becomes plausible given the environmental conditions in which the samples have been collected which are characterized by very scarce anthropogenic influence. As a general statement, the interpretation of the broad, unresolved NMR spectral features in the spectra of ambient organic aerosols remains challenging. It is a matter of fact that even in the organic aerosol analysis with very high-resolution analytical techniques like Orbitrap MS, the identification of chemical classes is prevalently based on given ranges of variability in the composition space (Van Krevelen plots) considered characteristic for some broad chemical classes (Jang et al., 2023). Even in the case of techniques like high-resolution MS capable to obtain thousands of chemical formulas, the number of isobaric isomers can be large and the chemical structures involved the most diverse ones. With respect to such techniques, NMR spectroscopy, in spite of the lower resolution, allows to confine the range of the major functionalities to given intervals of chemical shift. To illustrate the potential of NMR spectroscopy for molecular-level identification of organic tracers in the Antarctic aerosol samples, some examples of on the application of the Chenomx tool (Chenomx inc., evaluation version 9.0) are shown in the new Supplementary Figure S2 and S3

[Figure]

Figure S2. Example of identification of possible tracers using the extensive libraries of compounds offered by Chenomx NMR suite (Chenomx inc., evaluation version 9.0). In this figure are shown the expected NMR spectral patterns of some sugars and polyols, specifically sucrose (blue line), glucose (green line), glycerol (magenta line), D-threitol (brownish line) and lactate (orange line), against the NMR spectrum of PM1 sample S4 (black line). Sucrose and glucose molecular structures are also drawn in the figure, highlighting (with the red circles) the anomeric hydrogen used for their identification.

[Figure]

Figure S3. Another example, similar to previous figure, of identification of possible tracers using the extensive libraries of compounds offered by Chenomx NMR suite (Chenomx inc., evaluation version 9.0). Here it is reported an attempt of fitting the ambient PM1 spectrum of sample S4 with the signals expected for the molecules available in the database. Legend reports a list of compounds identified in this spectrum. Especially noteworthy are the signals of some fatty acids esters such as caproate, caprylate, suberate, sebacate, etc.

Connected to the previous comment, the authors should elaborate how they conclude from certain signals of groups (e.g. unsubstituted aliphatic groups, polysubstituted aliphatic groups, and anomeric and vinylic groups) to specific substances, such as lipopolysaccharide, low-molecular fatty acids or polyols.

Response: as already mentioned in the response to the previous comment (and now elaborated in the revised text and supplementary), we interpreted the spectra based on their characteristic patterns of resonances and chemical shifts, using for this scope libraries of reference spectra from the literature (of standard single compounds and/or mixtures from laboratory/chamber studies and or from ambient, in the field, at near-source stations). We also validated our interpretations using theoretical simulations of H-NMR spectra of atmospheric relevant molecules and extensive libraries of biogenic compounds offered by specific elaboration tools/software such as Chenomx NMR suite (Chenomx inc., evaluation version 9.0) and ACD/Labs (Advanced Chemistry Developments inc., version 12.01). More specifically, we discuss about "polysaccharides" or "sugars" because of the presence (unequivocally identified) of some molecular tracers such as glucose and sucrose, but also because of the concomitant presence of unidentified signal in alcoxy and anomeric region of the spectra (bands between 3.2-4.3ppm + peaks in the range 5-6ppm, typical of sugars); or "low-molecular weight fatty acids" because of the characteristic shape of bands at 0.9, 1.3, and 1.6 ppm (resembling a series of possible aliphatic chains of fatty acid esters such as caproate, caprylate, azelate, suberate, sebacate etc., even if it is not possible to identify all of them). And on the same criterium we made the hypothesis of the possible interrelationship between these polysaccharides and the fatty acids chains as coming from common lipo-polysaccharides precursors emitted/released by the marine biota.

Please revise the use of the abbreviations throughout the manuscript: (I) The authors introduce many abbreviations and are partially not needed, since they don't appear another time: e.g. 'open ocean (OO)' (Line 77). (II) The authors introduce other abbreviations repeatedly throughout the manuscript: (e.g. 'methanesulfonic acid (MSA)' (Line 46), 'methane-sulfonic Acid (MSA)' (Line 152), 'methanesulphonic acid (MSA)' (Line 290)). (III) The authors introduce abbreviations and do not use them consequently (E.g. Line 129: 'factor of 2 was used to estimate the WSOM from organic carbon…', 'organic carbon'=WSOC?, or DMA, TMA versus dimethyl amine, trimethyl amine)

Response: we thank the Reviewer for noticing these discrepancies. We revised the use of abbreviations along the whole manuscript.

It is not clear to me, which data (from chromatographic and H-NMR analysis) eventually were included in the factor analysis. Which signals were used to receive these five Factors?

Response: as mentioned both in section 2.4 of the main text and even more in details in the supplementary section 2, we applied non-negative factor analysis directly on the H-NMR spectra. The input variables are the spectral signals at the different chemical shift (binned every 0.02ppm) integrated and normalized in order to be proportional to the WSOC concentrations of each sample (always as reconstructed by H-NMR spectra following the approach based on functional groups distribution described in section 2.2).

In any case, in order to enhance clarity of the study we added a sentence to Section 2.4 of the main text:

The factor analysis was applied directly on the collection of spectra, using as input variable the spectral signals at the different chemical shifts (after several preprocessing steps described more in details in Supplementary Section S2).

Given the significance of the 'sympagic' versus 'pelagic' environment, particularly emphasized in the Results and Discussion sections, providing a summary of the current understanding of their influence on chemical composition from latest literatures in the introduction would greatly enhance the comprehensibility of the manuscript.

Response: we thank the Reviewer for appreciating the significance of the new concept proposed. The current paper greatly supports previous studies, showing the importance of different marine bioregions. We addressed the comment adding/editing the following paragraph in the Introduction:

"We previously showed that the microbiota of sea ice and the sea ice-influenced ocean (sympagic environment) can be a stronger source of atmospheric primary and secondary organic nitrogen (ON), specifically low molecular weight alkylamines (Dall'Osto et al., 2017; 2019) relative to open ocean areas not influenced by sea ice (pelagic ocean). Rinaldi et al. (2020) reported that non-methanesulfonic acid Water-Soluble Organic Matter (non-MSA WSOM) represents 6−8% and 11−22% of the aerosol PM1 mass originated in open ocean and sea ice regions, respectively. This study showed that the Weddell sea areas covered by open or consolidated packed sea ice (sympagic environment) is a strong source of organic nitrogen in the aerosol. Organic nitrogen compounds should be considered when assessing secondary aerosol formation processes in Antarctica beside the known role played by sulfur aerosols (Brean et al., 2021). By means of chamber experiments simulating primary aerosol formation on site in the same area around Antarctic peninsula and Weddell sea, Decesari et al. (2020) has previously reported that the process of aerosolization enriches submicron primary marine particles with lipids and sugars while depleting them of amino acids (Decesari et al., 2020). From these experiments emerged that the potential impact of the sea ice (sympagic) planktonic ecosystem on aerosol composition were overlooked in past studies, and that multiple eco-regions (sympagic environments, pelagic waters, coastal/terrestrial ecosystems) act as distinct aerosol sources around Antarctica (Decesari et al., 2020; Rinaldi et al., 2020). In particular, Decesari et al. (2020) found at least three main bioregions sources of water-soluble organic carbon (WSOC): (1) open Southern Ocean pelagic environments dominated by primary Sea-Spray Aerosol (SSA) mainly constituted of lipids and polyols, (2) sympagic areas in the Weddell Sea, with secondary sulphur and nitrogen organic compounds and (3) terrestrial land vegetation coastal areas, traced by sucrose in the aerosol."

Data availability: Atmospheric data from the Antarctic and Southern Ocean are sparse and hence precious to the scientific community. The authors have not published any raw data of atmospheric concentrations in the current version of the main manuscript or supplement, which makes it impossible to reproduce, reuse or compare their results. For a more transparent research and a

reusability of field data for future projects, the authors are strongly encouraged to publish their atmospheric concentrations on a public repository, such as PANGAEA. The comment 'Data are available from the authors on request' (Lines 456-457) should not be accepted by scientific journals anymore.

Response: we agree with the Reviewer and we decided to publish an asset of the data reported in this manuscript on Zenodo Data public repository (doi:10.5281/zenodo.10663787).
The dataset includes:
- concentrations of Water Soluble Organic Carbon and main other ions as measured by TOC-Analyzer (Shimadzu TOC-5000A) and Ion-Chromatography (IC, Dionex);
-H-NMR data in term of: the functional groups distribution; the concentrations of some molecular tracers (namely MSA, DMA and TMA); the contributions of the factors resulting from the Factor analyses of H-NMR spectral dataset;
-H-NMR ambient spectra at full resolution and after the binning (input matrix for the non-negative factor analysis) as well as the resulting spectral profiles of the sources identified by the statistical analysis

**Specific comments:**

Line 21: You mention the 'non-negative factor analysis' in the abstract. However, this term does not appear anywhere in the manuscript anymore.

Response: we thank the Reviewer for highlighting this discrepancy. We removed the "non-negative" from the abstract. On the contrary, because we believe it is important to specify that the factor analysis used in the study is based on non-negative algorithms, we added "non-negative" when we mention the methods the first time in Section 2.4.

Line 81: Who is 'we'? please add citation of reference.

Response: we changed the sentence to clarify:

"it has been previously reported that the process of aerosolization enriches submicron primary marine particles with lipids and sugars while depleting them of amino acids (Decesari et al., 2020)."

Line 93: 'It is becoming clear that in order to address important research questions in the polar regions it is essential measuring at multiple stations with a strong international scientific cooperation (Dall´Osto et al., 2019; Schmale et al., 2021).' In general, I agree with the authors that international collaborations are essential for advancing in science. However, I don't believe this sentence is suitable for finishing an introduction of a scientific paper. Instead, I recommend to replace this sentence with one that is related to your scientific findings or atmospheric implications.

Response: we removed the sentence and we replaced it with a new one on the relevance of our study:

"Our findings highlight the heterogeneity of the Antarctic ecosystems and how this heterogeneity impacts also on the organic aerosol sources allowing also - for the first time - to report some unique insights on their space and time variability in this region of the world."

Line 105: '[...] with mean annual air temperature of 3.5 C and annual precipitation ranging from 350 to 700 mm, primarily as summer rain.' First, the authors should give this meteorological information for both stations (not only Signy). Second, it would be more interesting to give a meteorological overview just for the period relevant for the campaign, not for the entire year.

Response: following the reviewer suggestion, we added some meteorological information also about Halley station, focusing more on summer period (relevant for our campaign). Moreover, in the supplementary Table S2 are already summarized some other meteo data more specific of the sampling periods.

"BAS Halley VI station (75°36'0" S, 26°11'0" W) is located in coastal Antarctica, on the floating Brunt Ice Shelf about 20 km from the coast of the Weddell Sea. Temperatures at Halley rarely rise above 0°C although temperatures around -10°C are common on sunny summer days. Winds are predominantly from the east. Strong winds sometimes pick up the surface snow, reducing visibility to a few metres. A variety of measurements were made from the Clean Air Sector Laboratory (CASLab), which is located about 1 km south-east of the station (Jones et al., 2008). BAS Signy station at Signy Island (60°43'0" S, 45°38'0" W) is located in the South Orkney Islands (Maritime Antarctic) and is characterized by a cold oceanic climate, extremely windy, with mean annual air temperature of 3.5 C and annual precipitation ranging from 350 to 700 mm, primarily as summer rain. Summer air temperatures are generally positive (record maximum 19.8°C), although sudden falls in temperature can occur throughout the summer (-7°C has been recorded in January). Signy is also extremely windy, with prevailing westerly winds."

Line 107: How did you prewash and prebake the quartz fiber filters? Solvent? Temperature?
Response: filters are pre-washed with ultrapure water (in order to reduce the ions concentrations in the blanks) and prebaked at 800° for 1h (for volatilize possible organic contaminants). We added this information in the revised text.

Line 119: 'Aerosol offline measurements and H-NMR analysis'- Isn't H-NMR one of the 'offline measurements'?
Response: true, we wanted to highlight the NMR but can be misleading. For this reason, we removed "and H-NMR analysis" from the title of the subsection.

Line 122-123: The authors measured organic acids, such as acetate, formate, oxalate. Where they all below limit of detection? Or why do they not appear as part of your WSOM discussion?
Response: we introduced the chromatographic measurements of organic acids because they are routinely measured with the procedure used in our lab and in this study as well. However, eventually we do not consider them in the discussion for several reasons: first of all, we wanted to focus on the H-NMR description of WSOM (that is more comprehensive than a discussion on few single organic acids molecules); second, the concentrations of these organic acids were often below the limit of detection (LOD) or too low to be interpreted. At Signy acetate concentrations are above the LOD only in 3 samples, in Halley in 6 out of 8 but with very low values (representing in any case less than 3% of WSOM); formate is always above LOD in Signy but represents less than 1% of WSOM, oxalate is not detectable at all in Halley samples and in Signy is above the LOD only in 3 samples (representing in any case less than 1% of WSOM). The same applies for organic cations others than di- and tri-methyl amines: methyl-, ethyl- and di-ethyl- amines even if detectable in some samples are not discussed in this paper intended to be more focused on H-NMR. We have so chosen to show the total water-soluble PM1 mass using directly the more straightforward total WSOM mass as measured by TOC-analyzer, that includes of course also these acids and amines. Nevertheless, we have now reported the concentrations of also these compounds in the dataset published on the Mendley public repository.

Lines 124-127: The authors give here information on the chromatographic analysis of inorganic ions. However, on which column and how did you analyze amines?
Response: the column and system used are the same for inorganic and/or organic cations (essentially amines). That was already specified in the section 2.3 of the original version ("An IonPac CS16 3 × 250 mm Dionex separation column with gradient MSA elution"). The procedure to separate and quantify amines is described in previous publications (e.g. Sandrini et al., 2016, already cited in the original manuscript)

Line 209: 'The meteorological conditions are not statistically different […]' How did you test the statistical difference?

Response: we applied both a parametric test (t-test) and a non-parametric one (Mann-Whitney test), on the mean values corresponding to each sample sampling-time. Both of them agree that no statistically significant difference can be found between the two periods at Signy, with a confidence interval of 95%.

Line 265: Why do you mention 'threitol' as a possibility here? Is glycerol not so sure? Threitol is not discussed anywhere else within the manuscript, so you should elaborate it a bit. When glycerol and threitol are possible polyols, then why not arabitol or mannitol (tracers for fungi in aerosol particles)?

Response: Glycerol is very evident in most of the spectra (as showed in the example below, Figure R1, and in the new supplementary Figure S2). D-threitol (as well as its diastereomer erythritol) exhibits a pattern of resonances overlapping with complex systems of NMR signals within the broader H-C-O region but distinct with respect to the peaks of glycerol. The signals at 3.7 ppm of chemical shift in the spectra of the samples are therefore "tentatively attributed" to D-threitol (as specified in the text). Mannitol and arabitol exhibit distinct patterns of resonances which cannot be confused with those of glycerol and D-threitol in NMR spectroscopy at 600 MHz.

[Figure]

**Figure R1**: fitting between the expected signals for glycerol (magenta line), D-threitol (brownish line), erythritol (blue line) and the H-NMR spectrum of sample S4 using the Chenomx NMR suite (Chenomx inc., evaluation version 9.0). Red line is the fitting line using the sum of the possible molecules available in the database.

Line 266: Do you equate 'low-molecular-weight fatty acids' with 'lipids'? I think this is deceptive and misleading for the readers to expand from a small subgroup to a big diverse class of molecules. Instead, I recommend to stick to the most correct terms possible.

Response: we thank the Reviewer for his/her suggestion. We removed the sentence as it can be misleading. The NMR analysis of WSOC cannot be accurate in detecting fatty acids with respect to other lipids. As explained above, the chemical classes identified on the basis of patterns in the unresolved NMR resonances are mainly based on the comparison between the spectroscopic properties of the ambient samples with those of aerosol source types, like the bubble bursting aerosols produced during tank experiments. In such samples, the most characteristic fingerprint for materials

originating from lipids is the pattern of complex resonances at 0.9, 1.2 and 1.5 ppm of chemical shift which is characteristic of aliphatic compounds with a linear structure. Such chemical structure cannot form from the oxidation of other biogenic compounds like isoprene and monoterpenes and, in this environment, must be linked to the atmospheric cycling of fatty acids. We refer here to "low-molecular weight fatty acids" (LMW-FA) because the methylenic chains are short (probably up to C8 – C10 at most) as C16 - C18 fatty acids are not recovered in WSOC. We sometimes refer to "lipids" to acknowledge that the LMW-FA can be part of a broader family of compounds although our method allows to fingerprint mainly the alkanoic acids (identified as LMW-FA).

Line 275: Based on which observation do you assume that this POA is 'apparently being transported for thousands of kilometers and across the Antarctic continent.'?
Response: One of the most striking result of the sampling campaign is that we found the same OA component (with its specific primary chemical features linked to sea-spray aerosol) at both sites, i.e. at locations 2000 km away from each other (with Halley distant from the ocean at least 200 kilometers). The remoteness of Halley from the sea excludes that marine POA are formed locally and must be advected from lower latitudes i.e. from the SO. In Figure 7, we show that during January, when the SOA component progressively takes over the SOA fraction of the organic aerosol, but around the 18th Jan an event of transport of POA have impacted both stations. This suggests that the POA concentrations are largely influenced by the synoptic-scale weather systems rather than by local production. We have modified the sentence to acknowledge that we report a first evidence for this phenomenon which deserves more investigation:
"In summary, this POA seems to be a common component of the sea-spray OA associated to open ocean areas across a wide range of longitudes and can be transported for thousands of kilometers."

Lines 461-465/ Author contributions: The abbreviation 'M.R.' could stand for both coauthors 'Mara Russo' and 'Matteo Rinaldi'. Rethink your abbreviation system. Who is 'D.S.C.B.'? (In the author list there is only a 'David C.S. Beddows'. What were the contributions of the coauthors 'Roy M. Harrison' and 'Thomas Lachlan-Cope'? They were not mentioned in this section.
Response: we thank the Reviewer for arising these mistakes/lacks. We have corrected the discrepancies in the revised version.

Figure 1b: I guess the bluish colors represent sea ice. If so, please add a legend to the plots.
Response: yes, the bluish areas represent the sea-ice cover extension. We added the info to the caption. We have also added a legend for the sea-ice percentages.

Furthermore, I was wondering what about the role of shelf ice. Was this considered in the air mass history analysis and discussion in your study? In Figure 1b the Antarctic shelf ice regions are currently presented with the same coloring like open ocean, which might be misleading.
Response: We have shaded the sea ice around the land mass of Antarctica marked on the map to help avoid any misleading

Figures 3 and 5 and 6b: Clarify the abbreviations denoting functional groups in the figure caption and ensure the usage adheres to proper English conventions for chemical terms (e.g., 'alif.' versus 'aliphatic').
Response: We thank the Reviewer for his/her suggestion. We changed the abbreviations accordingly to the CAS Standard Abbreviations & Acronyms lists (https://www.cas.org/support/documentation/references/cas-standard-abbreviations). What is not common abbreviation was explicitly reported in the text describing the H-NMR functional groups and identified tracers (paragraph 2.3).

Figure 6a: Use the same sequence for the substances in the plot as you did in Figures 2 and 4.
Response: We thank the Reviewer for his/her suggestion. We changed the sequence in the figure.

Table S1: Remove the last two sentences ('Whilst the start and end […] fits the purpose of the work presented') from the table caption. The column 'Month of the study' is redundant in regard of column 3,4,6 and 7. I recommend to remove column "Month of the study". Instead applying a proper date-time format to the remaining columns (e.g. dd/mm/yyyy hh:min)
Response: We accepted the suggestions and revised the Table accordingly

Table S2: Add Std to (S5;RH(%)). Should wind directions (WD) be averaged over sampling time considering that it is an angular dimension? I instead recommend to define four sectors (e.g. North: 315°-45°; East:45°-135°, South:135°-225°, West:225°-315°) to give in this table the percentage of time where wind came from which sector.
Response: The missing Std of RH has been added. Furthermore, about the wind direction we want to specify that the values reported in the original Table S2 are vector averages, i.e., considering already the angular dimension of wind direction. Anyway, we now calculated as suggested the percentage times where the wind came from each sector, whether N, E, S or W. These indeed corroborate our original vector average of wind direction and speed but are probably more straightforward for the reader (as suggested by the Referee) and so we replaced our vector average values with these time percentages in the revised Table S2

SI, Lines 60: How is betaine related to an oligomer?
Response: we thank the Reviewer for arising this oversight. We intended to use the term "osmolytes" and we changed it in the text

Si, Line 77 and in other parts of the manuscript: 'Dall'Osto et al. 2023, in prep.'. I would recommend to remove the year. When this manuscript is still in preparation and not submitted at least, it certainly won't be citable for 2023 anymore.
Response: we thank the Reviewer for arising this oversight. Based on the journal rules we actually have to remove all the *in prep.* references. We replaced that reference with another one already published (Dall'Osto et al., 2022a)

Figure S3: You identify glucose and sucrose in Factor 1, which are known to be neutral sugars. In Factor 5 you attribute a complete different chemical shift to 'neutral sugars', which seem to be identical to 'polysaccharides' and 'glycerol' in Factor 1. It appears inconsistent. Is it possible that the signals currently assigned to glucose and sucrose could also be other monosaccharides, disaccharides or derivatives, such as fructose, arabinose, trehalose or levoglucosan? How does your assignment of substances (e.g. sugars) match with the findings of other groups? Are there publications on glucose, saccharose, 'generic polysaccharides' in the atmosphere of the Antarctic or Southern Ocean (maybe using other analytical tools)?
Response: we thank the Reviewer for his/her attention and suggestions. We have already partially responded to this comment in the previous ones about the interpretation of spectral signals in H-NMR, but we complete the response answering to the specific questions raised here. The questions here involve two main aspects: 1. The different definitions of categories of compounds and 2. The unequivocal identification of single molecular compounds. About the first point, we sometimes used different definitions in different parts of the discussion mainly to highlight different aspects of the analyzed OA chemical composition. We used "neutral sugars" in opposition to "acidic sugars" (when we describe the main features of Halley samples) because these two categories have different prevalent signals in the H-NMR spectra: the former, characterizing more the range 3.2-4ppm, while the latter showing chemical shifts above 4ppm. The term "neutral sugars" is used in this part of the discussion just in contrast with the "acidic sugars", but as a matter of fact "neutral sugars" encompass

both saccharides than polyols like glycerol: this is now clarified in Table S5. While the signals above 4ppm (characteristic of Halley samples) can not be considered of the same category. So, we speculate on which other possible molecules they can represent (new supplementary figure S10 and S17 and related text).

About the second aspect, as already mentioned in previous replies, there are some signals that are clearly and unequivocally attributable to specific molecules: two of these are for instance the Sucrose and the Glucose anomeric doublets at 5.40 and 5.22ppm. These are clearly reported also in previous literature about Antarctic aerosol, sea-water and ice samples (Pautler et al., 2012; Dall'Osto et al., 2022b; Decesari et al., 2020). At the same time there are some molecules that can be excluded because should have specific signals in the spectra that are missing: this is the case of levoglucosan (characterized by a singlet at 5.45ppm, Paglione et al., 2014b) or arabinose (with a characteristic doublet at 4.5ppm and others around 3.9-4ppm, all missing) or trehalose (with its characteristic doublet at 5.18ppm). Instead fructose as well as other possible saccharides or polyols already cited (e.g., arabitol, mannitol, glycerol, etc.) can not be excluded and indeed they could be among those that make up the signal bands between 3.4 and 4ppm. We report here below an elaboration of the spectra using Chenomx evaluation tool to show the expected signals for some of these molecules. However, we cannot identify unequivocally them because if present they have very low concentrations and their specific peaks are all mixed together (scarce sensibility of the NMR for isolation of these compounds). And for this reason we just talk of generic "saccharides" or "neutral sugars".

[Figure]

Figure R2: fitting between the expected signals for some sugars relevant for atmospheric aerosol and the H-NMR spectrum of sample S4 using the Chenomx NMR suite (Chenomx inc., evaluation version 9.0). Expected signals in the alcoxy region (3.2-4.2ppm) of the spectrum are reported for: glycerol (magenta line), arabitol (or arabinitol, grey line), fructose (green line), threalose (blue line) and Manitol (red dashed line, because not fitting well).

Figure S8: Why do you differentiate between 'open ocean (<60°N)' and 'open ocean (>60°N)'. How would it impact the results of the measurements? Shouldn't it be '°S' instead of '°N'?

Response: we thank the Reviewer for arising this oversight. Of course, it is '°S' and not '°N', and the symbols for greater or smaller values are inverted. We modified the previous Figure S8 in the new S15, adding some more info in the caption. The distinction between latitudes higher or lower than 60°S is a proxy of the sea ice influence: all the latitudes greater than 60° are considered influenced by sea ice in a not negligible amount even if lower than sea ice marginal zone (that is for definition 15-85% covered by ice). To highlight the importance of the sea ice influence in the definition of this category we modified also its name in "Sympagic waters".

Figure S10: Eliminate the red underlining beneath 'Lac' (It appears the figure may have been copied from PowerPoint or Word).
Response: done.

Additional references:

Brege, M., Paglione, M., Gilardoni, S., Decesari, S., Facchini, M. C., and Mazzoleni, L. R.: Molecular insights on aging and aqueous-phase processing from ambient biomass burning emissions-influenced Po Valley fog and aerosol, Atmos. Chem. Phys., 18, 13197–13214, https://doi.org/10.5194/acp-18-13197-2018, 2018.

Chalbot, M.C., and Kavouras, I.G.: Nuclear magnetic resonance spectroscopy for determining the functional content of organic aerosols: a review; Environ Pollut. 2014, Aug:191:232-49. doi: 10.1016/j.envpol.2014.04.034.

Cleveland, M.J., L.D. Ziemba, R.J. Griffin, J.E. Dibb, C.H. Anderson, B. Lefer, B. Rappenglück Characterisation of urban aerosol using aerosol mass spectrometry and proton nuclear magnetic resonance spectroscopy, Atmos. Environ., 54 (2012), pp. 511-518, 10.1016/j.atmosenv.2012.02.074

Decesari, S., Paglione, M., Mazzanti, A., Tagliavini, E.: NMR spectroscopic applications to atmospheric organic aerosol analysis – Part 1: A critical review of data source and analysis, potentialities and limitations, Trends in Analytical Chemistry (TrAC),Vol 171, 117516, https://doi.org/10.1016/j.trac.2023.117516, 2024.

Gilardoni, S., Massoli, P., Paglione, M., Giulianelli, L., Carbone, C., Rinaldi, M., Decesari, S., Sandrini, S., Costabile, F., Gobbi, G. P., Pietrogrande, M. C., Visentin, M., Scotto, F., Fuzzi, S., and Facchini, M. C.: Direct observation of aqueous secondary organic aerosol from biomass burning emissions, P. Natl. Acad. Sci. USA, 113, 10013–10018, 2016.

Graham B, Mayol-Bracero OL, Guyon P, Roberts GC, Decesari S, Facchini MC, Artaxo P, Maenhaut W, Koll P, Andreae MO (2002) J Geophys Res 107(D20):8047

Liu, J., Dedrick, J., Russell, L. M., Senum, G. I., Uin, J., Kuang, C., Springston, S. R., Leaitch, W. R., Aiken, A. C., and Lubin, D.: High summertime aerosol organic functional group concentrations from marine and seabird sources at Ross Island, Antarctica, during AWARE, Atmos. Chem. Phys., 18, 8571– 8587, https://doi.org/10.5194/acp-18-8571-2018, 2018.

Pautler, B. G., Woods, G. C., Dubnick, A., Simpson, A. J., Sharp, M. J., Fitzsimons, S. J., and Simpson, M. J.: Molecular characterization of dissolved organic matter in glacial ice: coupling natural abundance 1H NMR and fluorescence spectroscopy, Environ. Sci. Technol., 46, 3753–3761, 2012.

Schkolnik, G., Rudich, Y. Detection and quantification of levoglucosan in atmospheric aerosols: a review. Anal Bioanal Chem 385, 26–33 (2006). https://doi.org/10.1007/s00216-005-0168-5

Suzuki, Y., Kawakami, M., and Akasaka, K.: 1H NMR application for characterizing water-soluble organic compounds in urban atmospheric particles, Environ. Sci. Technol., 35, 2656–2664, 2001.

---

## Author Comment (AC2)

We thank the Reviewer #2 for the helpful and constructive comments, which have led to significant improvements in the manuscript. We have carefully revised the text. Our point-by-point replies are given below (blue), following the referees' comments (black). Changes to the manuscript are marked with green. In our replies, the line numbers (when reported) refer to the revised manuscript.

**Reviewer 2**:
The authors present the analysis of filter data from two stations located at different latitudes in the Western Antarctic sector. They use ion chromatography and H-NMR to identify ionic species and water-soluble organic carbon functionalities and then apply positive matrix factorisation to the data to get insights into the sources of aerosols at those two sites. My major questions/comments are related to H-NMR and PMF. I am aware that authors from Bologna have previously used H-NMR for the analysis of aerosols, but this is not a method that is widely adopted in aerosol science and therefore, I think that this manuscript requires more information related to confidence in assigning certain functionalities (and especially compounds – e.g. lactic acid associated with Factor2) and also more on quantification (how is the intensity of various shifts converted to H amount and from that to water-soluble organic aerosol?) The authors are using WSOM (And later WSOA), when it should actually be WSOC. In relation to PMF I wonder if the small number of samples (n=22) is affecting the reliability of the method? I believe this should be briefly discussed in the paper. Have the authors considered downscaling and then including nss-SO4 and Na+ as a marker for seasalt (and possibly other ions) in the PMF? Or at least look at how some of the factors correlate with certain ionic compounds (e.g. POA pelagic with Na+).

Response: We appreciate the general positive feedback on the manuscript relevance and we thank the Reviewer#2 for highlighting the lack of clarity in the presentation of our observations and scientific conclusions. Based on his/her comments and suggestions (some of them shared also with Reviewer #1) we have revised the manuscript adding more information on H-NMR methodology for organic aerosol characterization both in term of functional groups distribution and in quantification of specific molecular tracers. As requested, we have also added information on the non-negative factor analysis techniques applied (of which PMF is only one method) on the series of H-NMR spectra. Replies to all the (several) specific questions raised by the Reviewer also in this general comment are detailed below. In general, however, first we want to clarify an important point: we recognize that the H-NMR complexity (together with other inherent limitations of the technique, like the poor sensitivity) still limit its use for atmospheric aerosol characterization and source apportionment with respect to other techniques, but we also would like to specify that the employ of NMR spectroscopy for characterization of atmospheric organic matrices is not new and actually now exists a quite large literature targeting the topic, which was also recently reviewed in a dedicated comprehensive review-paper just issued (Decesari et al., 2024). The NMR methodologies discussed in the present manuscript leverage 25 years of research in the field: they are not simply based on internal protocols of our laboratories of CNR-ISAC in Bologna.

Then, to reply point-by-point to the raised unclear definitions/explanations:

- regarding the functional groups and the tracers identified and quantified by H-NMR and presented/discussed in several figures and parts of the text, we have added new Tables (Table S4 and S5, also showed below) reporting a list and a description of the main chemical species/categories identified/measured by H-NMR. We also integrated with more info the text in section 2.3 as follow: "The main functional groups identified include unfunctionalized alkyls (H-C), i.e. methyls (CH3), methylenes (CH2), and methynes (CH) groups of unsubstituted aliphatic chains (i.e., also named later "Aliphatic chains"); aliphatic protons adjacent to unsaturated groups (benzyls and acyls: H-C-C=) and/or heteroatoms (amines, sulfonates: H-C-X, with X≠O), like alkenes (allylic protons), carbonyl

[revised manuscript text omitted]

| name of the species/ functional group*/ category of compounds** | ID of the species/ functional group | chemical shifts used for identification & quantification | examples for molecules | possible origin/source | references |
|---|---|---|---|---|---|
| *aromatic protons* | Ar-H | band 6.5-8.5 ppm | phenols, nitro-phenols [...] | biomass burning, [...] | Decesari et al., 2001; Tagliavini 2006; Decesari et al., 2007; Chalbot and Kavouras, 2014 |
| *anomeric and/or vinyl protons* | O-CH-O | band 6-6.5 ppm | vinylic protons of not completely oxidized isoprene and terpenes derivatives, of products of aromatic-rings opening (e.g., maleic acid), or anomeric protons of sugars derivatives (glucose, sucrose, levoglucosan, glucuronic acid, etc.) | biogenic marine mostly primary | Decesari et al., 2001; Schkolnik & Rudich, 2005; Tagliavini 2006; Decesari et al., 2007; Chalbot and Kavouras, 2014 |
| *hydroxyl/alkoxy groups* | H-C-O | band 3.2-4.5 ppm | aliphatic alcohols, polyhols, saccharides, ethers, and esters | biogenic marine primary | Chalbot and Kavouras, 2014 |
| *benzyls and acyls/ amines, sulfonates* | H-C-C= / H-C-X (X≠O) | band 1.8-3.2 ppm | protons bound to aliphatic carbon atoms adjacent to unsaturated groups like alkenes (allylic protons), carbonyl or imino groups (heteroallylic protons) or aromatic rings (benzylic protons) | biogenic/anthropogenic mostly secondary | Decesari et al., 2001; Graham et al., 2002; Decesari et al., 2007; Chalbot and Kavouras, 2014 |
| *unfunctionalized alkylic protons* | H-C | band 0.5-1.8 ppm | methyls (CH3), methylenes (CH2), and methynes (CH) groups of several possible molecules: fatty acids chains, alkylic portion of biogenic terpenes, etc. | biogenic/anthropogenic primary/secondary | Decesari et al., 2001; Graham et al., 2002; Decesari et al., 2007; Chalbot and Kavouras, 2014 |
| hydroxymethansulfopnic acid | HMSA | singlet at 4.39 ppm | | anthropogenic secondary | Suzuki et al., 2001; Gilardoni et al., 2016; Brege et al 2018 |
| methane-sufonate | MSA | singlet at 2.80 ppm | | biogenic marine secondary | Suzuki et al., 2001; Facchini et al., 2008a; Decesari et al., 2020 |
| di-methylamine | DMA | singlet at 2.72 ppm | | biogenic marine secondary | Suzuki et al., 2001; Facchini et al., 2008a |
| tri-methylamine | TMA | singlet at 2.89 ppm | | biogenic marine secondary | Suzuki et al., 2001; Facchini et al., 2008a |
| *N-osmolytes* | | singlets between 3.1 and 3.3 | betaine, choline and other structurally similar N-containing compounds not unequivocally identified (e.g., phosphocholine) | biogenic marine primary | Cleveland et al., 2012; Chalbot et al., 2013; Decesari et al., 2020; Dall'Osto et al., 2022b |
| betaine | Bet | singlet at 3.25 ppm (not quantified here but possibly quantifiable) | | biogenic marine primary | Cleveland et al., 2012; Chalbot et al., 2013; Dall'Osto et al., 2022b |
| choline | Cho | singlet at 3.18 ppm (not quantified here but possibly quantifiable) | | biogenic marine primary | Cleveland et al., 2012; Chalbot et al., 2013; Decesari et al., 2020; Dall'Osto et al., 2022b |
| *saccharides* | Sac | used synonymously for compounds carrying H-C-O groups in unresolved mixtures but when also anomeric protons (O-CH-O) are present | glucose, sucrose and other sugars structurally similar not unequivocally identified | biogenic marine primary | Graham et al., 2002; Facchini et al., 2008b; Decesari et al., 2011; Decesari et al, 2020; Liu et al., 2018; Dall'osto et al., 2022a |
| glucose | Gls | anomeric doublet at 5.22 ppm & specific structures between 3.5 and 4.2 ppm (not quantified but possibly quantifiable @5.22 ppm) | | biogenic marine primary | Decesari et al., 2020; Dall'Osto et al., 2022b |
| sucrose | Suc | anomeric doublet at 5.40 ppm & specific structures between 3.5 and 4.2 ppm (not quantified but possibly quantifiable @5.40 ppm) | | biogenic marine primary | Decesari et al., 2020; Dall'Osto et al., 2022b |
| *polyols* | | unresolved mixture not quantified (including glycerol and D-threitol) | glycerol, threitol, erytritol and structurally similar molecules not unequivocally identified | | |
| glycerol | Gly | specific structures at 3.55, 3.66 & 3.77 ppm (not quantified but possibly quantifiable @ 3.55 ppm) | | biogenic marine primary | Decesari et al., 2020; Dall'Osto et al., 2022b |
| D-threitol | | specific structures between 3.6 - 3.7 ppm (not quantified) | | biogenic marine primary | suggested in this study (to be confirmed) |
| *acidic-sugars / sulfonate esters* | | band 4-4.3 ppm (not quantified) | uronic acids, sulfonate-derivatives of polyols | biogenic marine primary/secondary | suggested in this study (to be confirmed) |
| *neutralsugars (saccharides) and polyols* | | band 3.5-3.9 ppm (not quantified) | glucose, sucrose and other sugars structurally similar not unequivocally identified | biogenic marine primary | Graham et al., 2002; Facchini et al., 2008b; Decesari et al., 2011; Decesari et al, 2020; Liu et al., 2018; Dall'osto et al., 2022a |
| *low-molecular weight fatty acids or "lipids"* | LMW-FA | unresolved complex resonances at 0.9, 1.3, and 1.6 ppm in the H-C spectral region | fatty acids (free or bound) from degraded/oxidized lipids (e.g. caproate, caprylate, suberate, sebacate, etc.) and similar compounds owning a chemical structures of alkanoic acids. | biogenic marine primary | Graham et al., 2002; Facchini et al., 2008b; Decesari et al., 2011; Decesari et al, 2020; Liu et al., 2018 |
| lactic acid | Lac | doublet 1.37-1.36 ppm & quadruplet at 4.23 ppm (not quantified but possibly quantifiable @1.37-1.36 ppm) | | biogenic marine primary | Suzuki et al., 2001; Decesari et al., 2020; Dall'Osto et al., 2022a |

-regarding the use of WSOM and WSOC, we used WSOM in mass budget calculations (i.e., together with inorganic ions mass concentrations) while we used WSOC in carbon mass budget calculations. Carbon units are used for NMR functional group budget as well as for NMR factor concentrations. It should be noted that WSOM and WSOC are based on the same measurement of total carbon in the sample water extracts (using the TOC analyser), so that: WSOM = (non-MSA WSOM) + MSA; where (non-MSA WSOM) = 2* (non-MSA WSOC) + (MSA WSOC); with (MSA WSOC) = 12/96*MSA representing the carbon concentration associated with MSA. We have better explained this in Section 2.3. WSOA refers to the organic mass in the aerosol and its apportionment in different "types". So, it is substantially used as a synonym of WSOM but when we start to discuss the organic aerosol sources/components (also separated in primary and secondary OAs or POA and SOA, etc.).

- regarding PMF and non-negative factor analysis, we have added a more comprehensive description of the methodology (preparation of the input data, choice of the best number of factors, analysis of the residuals, interpretation of the resulting factors, sensitivity tests on the robustness of the solution, etc.) in the Supplementary Section S2.

About the limited number of samples used, we would like to comment here that the definition of a minimum dataset size in factor analysis is an issue not yet resolved univocally: even if there are a lot of different theoretical rules (Arrindell & van der Ende (1985), Velicer & Fava (1998), and MacCallum et al. (1999) have reviewed many of these recommendations), many studies have demonstrated that the general rules of thumb of the minimum sample size are not always valid and useful (MacCallum et al., 1999; Preacher & MacCallum, 2002). The minimum level of N (dataset size) was object of a very high number of studies that demonstrated its dependency on other aspects of design, such as: the communality of the variables (percent of variance in a given variable explained by all the factors jointly and interpreted as the reliability of the indicator) (Hogarty et al., 2005; MacCallum et al., 2001; Costello & Osborne, 2005); the degree of overdetermination of the factor (or number of factors/number of variables) (Preacher & MacCallum, 2002; MacCallum et al., 1999); the internal variability of the dataset (Paglione et al., 2014b); etc.
The present study is based on a short timeline of samples (# = 22), but the variability in NMR composition was high, and this is witnessed by the poor correlations between the spectral profiles (except between the Factor 3 and 5, which have strong MSA and DMA peaks in common, even if alxoxy region very different) showed in new supplementary Figure S5. In turn, the variability in the chemical composition was influenced by the great variability between the two sites and the air mass origins influencing them.

For the above reasons, we have chosen the most appropriate number of factors based on the comparison between the outcomes of distinct factor analysis methods. The best agreement was achieved by far for the 5-factor solution, and this remains the most realistic NMR spectral deconvolution in this experiment.
We have also checked the robustness of the chosen solution running the models with different-sized input matrixes (reducing the number of variables, eliminating MSA signal at 2.80ppm from the NMR spectra, for instance, or alternatively the number of samples, eliminating some samples from the time series, such as S3 or H5 for instance). A comparison between the results of all these different elaborations is showed in original supplementary Figure S4, now Figure S11 (and further explained/discussed in Section S2).

-about the possibility to add nss-SO4 and Na to the factor analysis input matrix, we don't believe it can add any additional information except the correlation (or anti-correlation) of those species. So, considering also the complexity of mixing NMR spectral data with concentrations of single species in a single PMF where hundreds of species are already included (mixing very different variables in PMF is possible, but never straightforward, as suggested by a wide body of literature, e.g. Slowik et

al., 2010), we actually prefer to use sea-salt and other components for external correlations with the NMR-factors (which contemporary also help in evaluate the validity and robustness of the NMR-factors interpretation). We have already commented about these correlations in the discussion of the original version, but to enhance the clarity of the results we decided to add also a correlation table between NMR factors and other chemical-species in the Supplementary (new Table S6)

The authors have not discussed MSA separately, but have grouped it into WSOM. As one of the most important compounds in marine environments, I think it should be reported separately. Looking at the MSA/nss-SO4 at those two sites would be valuable. There might be differences that can then be related to the origin of airmasses and atmospheric circulation. I also think that it would be valuable for the atmospheric community to include a supplement table containing ionic composition for all the samples.

Response: MSA has been considered part of the WSOM (as it is an organic molecule) in the graphs/paragraphs where only total WSOM was shown and commented (Figure 2 and 4), but it has been already considered separated in the WSOC speciation made by H-NMR in the original version (Figure 3, 5 and 6) and quantified as mass concentration in original Figure S2. For sake of clarity of the graphs in Figure 2 and 4, we have preferred in the original version to not show MSA separated. Anyway, we understand the concern of the Reviewer about the fact that an average OM:OC-ratio of 2 (used for the conversion of WSOC in WSOM as taken from the literature) is not representative of the MSA mass-to-carbon-ratio and that in an OA strongly dominated by MSA the average OM:OC ratio can be higher than 2. For this reason, we accepted the suggestion of the Reviewer to comment more about MSA proportion to the total WSOM and about MSA/nss-SO4 in the text (paragraph 3.1 and 3.2) and to separate MSA (in mass concentration) from WSOM also in the graphs of Figure 2 and 4. We also calculated a new total WSOM that is the sum of MSA mass concentration and the rest of WSOC multiplied by an average OM:OC-ratio of 2 (and we explained it in the revised text, section 2.3). For the same reason, also the average and standard deviation values discussed in the Section 3.1 and 3.2 result slightly changed in the revised version (not modifying the main messages, anyway).
To enhance accessibility of the data we also decided to publish an asset of the data reported in this manuscript on Zenodo Data public repository (doi:10.5281/zenodo.10663787).
The dataset includes:
- concentrations of Water Soluble Organic Carbon and main other ions as measured by TOC-Analyzer (Shimadzu TOC-5000A) and Ion-Chromatography (IC, Dionex);
-H-NMR data in term of: the functional groups distribution; the concentrations of some molecular tracers (namely MSA, DMA and TMA); the contributions of the factors resulting from the Factor analyses of H-NMR spectral dataset;
-H-NMR ambient spectra at full resolution and after the binning (input matrix for the non-negative factor analysis) as well as the resulting spectral profiles of the sources identified by the statistical analysis

There should be some consistency when reporting numbers. For average values the authors sometimes include error bars and in some cases they don't. Please include error bars in all of your reported average values.
Response: We have revised all the average values reported in the text and we have added the missing ± ranges (representing standard deviations). Error bars in the figures (where reported) represent in any case the inter-sample variability and are showed only if useful to transmit the message and not confusing it. For instance, error bars in figure 9 are considered appropriate only for the plots reporting the statistics for parallel sample sets (either Signy and Halley).

My other questions/comments are listed below.
Southern Ocean and Antarctica are very undersampled regions of the world and data coming from that part of the world is very valuable. However, I think that the authors should explore their data a

bit more (e.g. MSA, MSA/nss-SO4 ratio) and provide more details related to their analytical and data treatment approaches. I would love to see this manuscript published in the ACP, but only after major revisions.

Response: we accepted the suggestion of the Reviewer to comment more about MSA and MSA/nss-SO4 in the text (paragraphs 3.1 and 3.2). Moreover, we added in the Supplementary two new Tables reporting a list and a description of the main chemical species/categories measured by chromatographic and spectroscopic analyses used in the manuscript (new Table S3 and S5, also reported in Replies to Reviewer#1).

To enhance accessibility of the data we also decided to publish an asset of the data reported in this manuscript on Zenodo Data public repository (doi:10.5281/zenodo.10663787).

The dataset includes:

- concentrations of Water Soluble Organic Carbon and main other ions as measured by TOC-Analyzer (Shimadzu TOC-5000A) and Ion-Chromatography (IC, Dionex);

-H-NMR data in term of: the functional groups distribution; the concentrations of some molecular tracers (namely MSA, DMA and TMA); the contributions of the factors resulting from the Factor analyses of H-NMR spectral dataset;

-H-NMR ambient spectra at full resolution and after the binning (input matrix for the non-negative factor analysis) as well as the resulting spectral profiles of the sources identified by the statistical analysis

Specific comments/questions:

Line 20: "average 25-33%" seems a bit unusual. Is 25-33% minimum to maximum. It would be better to report it as average +/- ??. Also ~30% seems like a lot and I can see from reading further that this is due to including MSA in the WSOM. This is one of the most important compounds in the Antarctic environment and should be reported separately.

Response: we thank the Reviewer for highlighting the lack of clarity here: "average 25-33%" was not a range, but the averages of the two sites respectively. As already explained in a previous response we also agreed to separate MSA and so we calculated a new total WSOM mass, reporting in the revised abstract the new average values of the relative contribution of WSOM to the total water-soluble PM1 mass of 37 and 29%, for Signy and Halley respectively. We explained the calculations in Section 2.3 as follow:

"The water-soluble organic carbon (WSOC) content was quantified using a TOC thermal combustion analyzer (Shimadzu TOC-5000A). Given MSA high relative contribution to the total organic mass, it is separated by subtracting its carbon contribution (in $\mu gC\ m-3$) from total WSOC, obtaining the non-MSA WSOC. A carbon-to-mass conversion factor of 2 was used to estimate the non-MSA water-soluble organic matter (non-MSA WSOM) from non-MSA WSOC measurements, following the values suggested for marine organic aerosols by Jung al. (2020). The total WSOM was then calculated as the sum of MSA and the non-MSA WSOM mass concentrations."

Line 26: latitudinal gradients for which aerosol components?
Line 26: the last sentence is quite vague. Is it possible to be more specific? E.g. what are the main differences between pelagic and sympagic emissions?

Response: following the suggestions of the Reviewer, we revised the abstract to make it more informative and clearer.

Line 30: instead "considered a window to the preindustrial atmospheric condition", I suggest: considered to be a proxy for the preindustrial atmospheric conditions. Why is that so? I suggest adding a sentence providing an explanation of why is that region a proxy for the pre-industrial atmosphere?

Response: we accepted the Reviewer suggestions and changed the sentence as follow:

"Given their distance from major anthropogenic sources, the Southern Ocean (SO) and Antarctica are considered to be a proxy for the preindustrial atmospheric condition and processes…"

Line 33: $10^6$ instead of 106
Response: corrected in the revised version.

Line 39: waves breaking…and bubble bursting
Response: added in the revised version.

Line 42: if aerosol chemistry in southern high latitudes is described as "much more complex" then it would make sense to give more than one example confirming that complexity. (E.g. Involvement of iodine based compounds should be mentioned)
Response: added in the revised version: "Another potentially key component for new particle formation in Antarctica is iodine, which is known to form new particles via iodic acid nucleation (Saiz-Lopez et al., 2007; Baccarini et al 2021)."

Line 42: no need for capital s in Southern
Response: corrected in revised version.

Line 53: there are studies presenting different results (i.e. MSA representing a significant portion of organic mass) and they should be mentioned in the introduction. E.g.:
-Fossum, K.N., Ovadnevaite, J., Ceburnis, D. et al. Summertime Primary and Secondary Contributions to Southern Ocean Cloud Condensation Nuclei. Sci Rep 8, 13844 (2018). https://doi.org/10.1038/s41598-018-32047-4
-Matteo Rinaldi, Marco Paglione, Stefano Decesari, Roy M. Harrison, David C.S. Beddows, Jurgita Ovadnevaite, Darius Ceburnis, Colin D. O'Dowd, Rafel Simó, and Manuel Dall'Osto, Environmental Science & Technology 2020 54 (13), 7807-7817
-Jung, J., Hong, S.-B., Chen, M., Hur, J., Jiao, L., Lee, Y., Park, K., Hahm, D., Choi, J.-O., Yang, E. J., Park, J., Kim, T.-W., and Lee, S.: Characteristics of methanesulfonic acid, non-sea-salt sulfate and organic carbon aerosols over the Amundsen Sea, Antarctica, Atmos. Chem. Phys., 20, 5405–5424, https://doi.org/10.5194/acp-20-5405-2020, 2020.
Response: we do not downplay the importance of MSA, on the contrary  we do say that MSA is already recognized as the most common and among the most abundant organic components. Then we explain that much less is known about the organic part which is NOT MSA and which is also abundant in other studies (as well as in ours). In any case, trying to be clearer on the point we modified the text as follow: "As regards of secondary (gas-to-particle formation) aerosols, the DMS-derived nssSO42-(Charlson et al., 1987; Vallina et al., 2007) is normally accompanied by organic sulfur species, the best known and usually more abundant of which is methanesulfonic acid (MSA) (Rankin and Wolff, 2003; Legrand et al., 2017b; Fossum et al., 2018)."
Of the suggested additional references, we added here only Fossum et al., 2018 because is the one more supporting the statement and because the others are already cited in other sections of the text.

Line 62: I suggest "airmass origin" instead of airmass type
Response: modified in revised version.

Line 64: what do you mean by continental – Australian or Antarctic?
Response: we substantially reported the classification by Humphries et al. (2021). Overall "continental" means from inhabited areas and is linked to anthropogenic, indeed.

Line 65: latitudinal gradient in CCN was also reported in Humphries et al (2021), not only Sanches et al (2021). Please include that in the sentence.

Response: included in revised version.

Line 70: should be >60 S
Response: corrected in revised version. Thanks for noticing the misspelled value.

Line 72: By analyzing simultaneous aerosol size distribution measurements at three sites, Lachlan-Cope et al (2020) showed..
Response: modified in revised version.

Line 73: is more complex instead of are more complex
Response: modified in revised version.

Line 73: it might not be understandable to every reader what the authors mean by "the simple sulfate-sea salt binary combination" and how does that reflect on aerosol concentrations and size distribution. I suggest revision of this part to make it clearer.
Response: modified in revised version as "more complex than the simplified view of particles as composed by the sulfate–sea-spray combination,"

Line 83: Sentence starting with "Overall.." should be split into two separate sentences: Multiple eco-regions around Antarctica were found to act as distinct aerosol sources (Decesari et al., 2020; Rinaldi et al., 2020). However, the potential impact of the sea ice (sympagic) planktonic ecosystem on aerosol composition is frequently overlooked.
Response: the meaning of the sentence wanted to be different from what the Reviewer got from it. So, trying to be clearer, we modified it as follow: "From these experiments emerged that the potential impact of the sea ice (sympagic) planktonic ecosystem on aerosol composition were overlooked in past studies, and that multiple eco-regions (sympagic environments, pelagic waters, coastal/terrestrial ecosystems) act as distinct aerosol sources around Antarctica (Decesari et al., 2020; Rinaldi et al., 2020)."

Line 84: The sentence starting with "Decesari et al.." needs to be revised to provide context. E.g. Decesari et al have investigated aerosol composition in…(where?) and found …
Response: considering that replying also to Referee#1 we changed the paragraph citing Decesari et al. (2020) just before in the context of a well-defined study, we believe it is redundant to specify again here location and context of their measurements. So, we revised the paragraph as follow: "By means of chamber experiments simulating primary aerosol formation on site in the same area around Antarctic peninsula and Weddell sea, Decesari et al. (2020) has previously reported that the process of aerosolization enriches submicron primary marine particles with lipids and sugars while depleting them of amino acids (Decesari et al., 2020). From these experiments emerged that the potential impact of the sea ice (sympagic) planktonic ecosystem on aerosol composition were overlooked in past studies, and that multiple eco-regions (sympagic environments, pelagic waters, coastal/terrestrial ecosystems) act as distinct aerosol sources around Antarctica (Decesari et al., 2020; Rinaldi et al., 2020). In particular, Decesari et al. (2020) found…"

Line 105: why is annual temp and precipitation reported for Signy and not for Halley? Should be rerported for both, or none.
Response: following the reviewer suggestion, we added some meteorological information also about Halley station, focusing more on summer period (relevant for our campaign). Moreover, in the supplementary Table S2 are already summarized some other meteo data more specific of the sampling periods.

"BAS Halley VI station (75°36'0" S, 26°11'0" W) is located in coastal Antarctica, on the floating Brunt Ice Shelf about 20 km from the coast of the Weddell Sea. Temperatures at Halley rarely rise above 0°C although temperatures around -10°C are common on sunny summer days. Winds are predominantly from the east. Strong winds sometimes pick up the surface snow, reducing visibility to a few metres. A variety of measurements were made from the Clean Air Sector Laboratory (CASLab), which is located about 1 km south-east of the station (Jones et al., 2008). BAS Signy station at Signy Island (60°43'0" S, 45°38'0" W) is located in the South Orkney Islands (Maritime Antarctic) and is characterized by a cold oceanic climate, extremely windy, with mean annual air temperature of 3.5 C and annual precipitation ranging from 350 to 700 mm, primarily as summer rain. Summer air temperatures are generally positive (record maximum 19.8°C), although sudden falls in temperature can occur throughout the summer (-7°C has been recorded in January). Signy is also extremely windy, with prevailing westerly winds."

Line108: are the sampling sites influenced by station activities? How was ensured that aerosols coming from station combustion activities were not collected?
Response: Precautions have been taken to reduce the risk of sample contamination from the stations, such as (1) distancing the sampler from the base as much as possible, (2) locating the sampler so that it was not downwind to the station with respect to the dominant circulation and (3) exclusion of the potentially contaminated wind sector by an automatic wind direction driven trigger. Sample analysis supports the goodness of the above precautions as we did not detect in significant amount any anthropogenic tracers (e.g., aromatics, HMSA, levoglucosan, etc.) for which we know H-NMR can be very sensitive (Suzuki et al., 2001; Gilardoni et al., 2016; Brege et al 2018).

Line 123: what inorganic ions were measured? It should also be described here how sea salt was calculated.
Response: we thank the Reviewer for highlighting this flaw of clarity. Following his/her suggestion (as well as the Reviewer#1 ones) we added information on the ionic species measured and on the sea-salt calculation in Section 2.3 of the main text. We also added a new Supplementary Table reporting a list and a description of the main chemical species measured by ion chromatography (new Table S4, also reported below).

Line 137: Should it be DCOOD instead of HCOOH?
Response: no, it is non-deuterated formic acid. We usually used it because the deuterated version is much more expensive. It provides limited interference in the spectra because of the relatively small amounts required for buffering and especially because the NMR peak of HCOOH is located quite far away from most of the resonances found in the NMR spectra of the Antarctic samples.

Line 143: How are hydrogen concentrations calculated and how are they converted to organic carbon? This quantification should be described in a bit more detail.
Response: as already mentioned previously, regarding the functional groups and the tracers identified and quantified by H-NMR and presented/discussed in several figures and parts of the text, we have added a new Tables (Table S3 and S5) reporting a list and a description of the main chemical species/categories identified/measured by H-NMR. We also integrated with more info the text in section 2.3 as follow:
Organic hydrogen concentrations directly measured by H-NMR were converted to organic carbon. Stoichiometric H/C ratios were specifically assigned to functional groups using the same rationale described in previous works (Decesari et al., 2007; Tagliavini et al., 2006): briefly, the choice of specific H/C molar ratios is based on the expected stoichiometry and structural features of the molecules that every region of the H-NMR spectra can actually represents in atmospheric aerosol samples on average. The H/C ratios used in this study are showed in Supplementary Table S4.

Line 163: how does the low number of samples (22) affect the performance of factor analysis? Also, both methods should be briefly described in the Supplement.
Response: already replied above to previous general comments.

Line 190: as already mentioned, it should be described in the methodology how sea salt was calculated; 55% +/- ??
Response: already replied above to a previous comment.

Line 191: I am assuming MSA is included in WSOM? I suggest reporting it separately as it is a major DMS oxidation product.
Response: already replied above to a previous comment.

Line198: some percentages include standard deviation/error, some don't. please be consistent and include +/- with all of your reported values
Response: we have revised all the average values reported in the text and we added the missing standard deviations.

Line 203: with respect
Response: corrected, thanks.

Line 207: Enriched by mSA
Response: corrected, thanks.

Line 209: You could colour Signy backtrajectories in Figure 1 b with lighter and darker red with each colour representing one of the two described periods. Or Fig6 backtrajectories could be coloured by a date.
Response: we thank the Reviewer for the suggestion, but we think that adding other colors to the maps can reduce the clarity of the Figures not adding fundamental info. For this reason, we didn't accept the suggestion and we left the colors of the backtrajectories unchanged.

Line 217: lower variability is likely due to the short sampling period
Response: in principle this could be reasonable, but also considering just the parallel sampling period at both the sites, Signy shows a higher variability. Moreover, here in the results section we just reported the results of the analysis not commenting the reasons, which are instead treated more extensively later in the discussion.

Line 221: add +/- to your percentages
Response: we have revised all the average values reported in the text and we added the missing standard deviations.

Line 224: primary aerosol instead of primary sources.
Response: done.

Line 30: primary aerosols instead of primary sources.
Response: done

Line 232: how far is Halley from the open ocean? Halley is a coastal site, so please specify here what do you mean by open ocean.
Response: we thank the Reviewer for highlighting the misleading info given in different sections of the text: Halley was described as a coastal site in the methods (section 2.1) because it is actually not far from the continental coast, but it can not be considered close to the sea water (or open ocean

environments) because it is distant from the open sea hundreds of kilometers, depending on the extension of the ice pack covering the Weddell along the whole year (with a summer minimum of about 200 km in any case). We explained better the location of the site in the revised version of section 2.1:

"BAS Halley VI station (75°36'0" S, 26°11'0" W) is located in coastal Antarctica, on the floating Brunt Ice Shelf about 20 km from the coast of the Weddell Sea, but distant hundreds of kilometers from the open ocean (at a variable distance along the year depending on the extension of the pack ice and floating sea-ice covering the Weddell in the different seasons, approximately 200km during summer)."

Line 233: contrary to Signy where alkyl-amines represent ??% of the PM1 mass.
Response: the corresponding value for Signy was added in the revised version.

Line 257: Please provide a sentence or two describing based on what was the 5 factor solution described as the most robust and informative.
Response: we added the following sentence to better explain our interpretation:
"Here we report a description of the 5-factors solution which was identified as the most robust and informative one (Figure 7) based on the best separation of interpretable spectral features and on the best agreement between the two algorithms applied with respect to both spectral profiles and contributions."

Line 258: how are CWTs calculated? In Fig 8, what do the units for the colour legend represent? CWT should be described in the methodology and colour scale description can be given in Fig 8 caption.
Response: As already explained in the original version, the CWT maps were calculated using backtrajectories calculated by HYSPLIT and the trajectory level plots belonging to the Openair package (Carslaw and Ropkins 2012), using default settings. In particular, the CWT approach considers the concentration of a species together with its residence time in a grid cell. Briefly, this procedure creates a concentration field from a grid domain to identify source areas of pollutants. To explain better the methodology we added a sentence to the revised version as follows:

"The CWT approach uses the concentration measured upon a trajectory's arrival at site and the residence time of that trajectory in each grid cell it passes through to create a mean concentration for each grid cell. When plotted as a map, this shows that air masses passing over which cells would, on average, give higher concentrations at the measurement site."

Moreover, we added a description of the color scale in the Fig.8 caption.

Line 262: this is Water soluble fraction. Lipids are not water soluble. Could alkyl chains be coming from something else?
Response: the reviewer is true: lipids are not water-soluble per definition. However, the H-NMR spectral profile described here is characterized by specific bands at 0.9, 1.3, and 1.6 ppm (resembling a series of possible aliphatic chains of fatty acid esters such as caproate, caprylate, azelate, suberate, sebacate etc.) which are interpreted as degradation products of lipids. Moreover, those signals are strongly related (i.e., concomitant presence in the spectra) with signals in the alcoxy and anomeric region associated to saccharides and polyols (among which some were identified at the molecular level, such as glucose, sucrose and glycerol). In previous studies focusing on bubble-bursting experiments in which the water-insoluble fraction was also characterized (Facchini et al., 2008b), the above NMR features of WSOC were associated with the presence of complex mixtures of lipids and sugars. Therefore, even if the present study focuses on WSOC, we interpret the presence of alkanoic

acids (like low-molecular weight fatty acids, LMW-FA) as a fingerprint for lipids, even if lipids per se are not detected directly but rather from degradation products of them. For these reasons we put "lipids" in the name of this factor in the original version to trace source information.

We agree with the Referee that in the original version we lacked of clarity, and for this reason as already mentioned in the response to Reviewer#1 we have added more info on interpretation and validation of the H-NMR signals in the revised text and supplementary (new Section 2.3; Table S5, Figure S2 and S3).

Line 278: what does "lac" stand for?

Response: we thank the Reviewer for noticing the lack of introduction of the abbreviation for the lactic acid. We left the "lac" in the name of the factor (between quote marks), introducing the abbreviation few lines later when we discuss about lactic acid.

Line 280: how can you be sure that peaks at 1.35 and 4.21 ppm belong to lactic acid? Based on figure S4 Factor 2 seems to be dominated by MSA and DMA. It looks like factors and their corresponding NMR spectra have been mixed up.

Response: we thank the Reviewer for noticing this discrepancy: it is actually a mistake in the Figure labelling: Factor2 in the original Figure S4 was showing the spectral profile of the Factor4 and viceversa. We corrected the figure in the revised version.

About the identification of lactic acid, its specific signals in H-NMR spectra are well known and reported in all the metabolome databases and identified in aerosol samples since the first publications using the H-NMR to characterize organic aerosol (e.g., Suzuki et al., 2001). We provide here below a comparison of the expected signals for lactate in the database of the Chenomx NMR suite (Chenomx inc., evaluation version 9.0) and the spectrum from one Signy ambient PM1 sample.

[Figure]

Figure R1: fitting between the expected signals for lactic acid (or lactate, orange line) and the H-NMR spectrum of sample S4 (black line) using the Chenomx NMR suite (Chenomx inc., evaluation version 9.0).

Line 286: How are backtrajectories shorter? Backtrajectories look the same for all factors, they are just coloured differently. Backtrajectories coloured by the highest intensity seem to be coming south from Signy

Response: we agree that the sentence can be misleading and we replaced it with the following one:

"This factor is found only in the first subperiod of Signy (samples S1-S5) and the associated CWT maps are similar but somewhat showing higher concentrations closer to the site with respect to Factor 1. For this reason, we consider it as a second marine POA factor influenced by aerosol sources in coastal/sympagic areas around the Antarctic Peninsula."

Line 291: is there a difference in NMR shift between different alkyl-amines (i.e. can you tell wth certainty that 2.71 is coming from DMA)?

Response: yes, of course there is a clear difference. As explained now better in the revised text (section 2.3) and in the new Table S5, we can unequivocally identify and quantify DMA and TMA by their singlets at 2.71 and 2.89ppm, respectively. Moreover, the quantification of these two alkyl-amines is further supported by the comparison with IC measurements for the same species (already reported in original Figure S2 and in the new S4).

297: planetary boundary layer instead of PBL. Indeed, the majority of backtrajectories at Halley are coming from the Antarctic continent, but that alone is not sufficient to describe katabatic outflow. Humidity and temperature need to be considered here as well. How was it determined what percentage of time was spent below or above the boundary layer? That should be outlined in the methodology.

Response: abbreviation is now explained, thanks. We also agreed with the Reviewer that because we don't have the data to demonstrate the occurrence of katabatic outflows, we removed the definition and discuss only of "free-tropospheric circulation" over the continent. Our intention was just to highlight that the air masses reaching Halley spent much more time above the PBL with respect to the ones reaching Signy, showing an higher level of processing for some components even when originating from similar environments/areas.

301: same as for DMA and MSA in the Factor 3, indicate TMA NMR shift here. DMA is also present here. To me it would make more sense to call this factor "alkyl-amines + MSA"

Response: it is explained now better in the revised text (section 2.3) and in the new Table S5. we can unequivocally identify and quantify MSA, DMA and TMA by their singlets at 2.80, 2.71 and 2.89ppm, respectively. Moreover, the quantification of MSA and of these two alkyl-amines is further supported by the comparison with IC measurements for the same species (already reported in original Figure S2).

309: HC-O and H-C seem to be very low comparing to MSA and DMA

Response: actually, in term of fractional abundance (i.e., percentage of the integrals of specific bands/signals to the total of the spectra) H-C-O and H-C represent 27 and 24% respectively of the commented spectral profile (Factor5), while MSA and DMA represent 22 and 10%, respectively. So, even if MSA and DMA singlets are much more evident in the graph reported in the Figures (because they are sharp singlets, easy to be recognized), our statement is still valid at a detailed analysis of the spectral profiles. For this reason, we didn't change it in the revised version.

313: have never been

Response: corrected, thanks.

314: this is the first time that the acidic nature of aerosols in Halley has been mentioned. Where is this coming from and how has this been demonstrated? This cannot be assumed only based on high abundance of nss-SO4. To me it looks like factor 5 is simply a mix of Factor 1 and 2 and in line with that I think there is no enough evidence to hypothesise about the formation of organo-sulfates.

Response: the hypothesized acidic nature of PM1 at Halley came from the ionic composition described in the text and in the figures, dominated by sulfate and without enough ammonium and/or other cations able to neutralized it. Acidity is a quite typical characteristic of fine marine aerosol (Fridlind and Jacobson, 2000; Keen et al., 2004; etc.) and quite evident from the graphs in Figure 5. So, we have not believed it was necessary to demonstrate it introducing thermodynamic calculations (even if actually we have also preliminarly applied ISORROPIA-II thermodynamic model on the ionic composition of all the samples) because the paper contains already a lot of data and information.

On the other hand, we use here this consideration about the possible acidic nature of aerosol at Halley just to speculate on the formation of possible organo-sulfates which could possibly explain the peculiar spectral features in H-NMR spectra of Halley samples (the signals at 4-4.5ppm, better explained later and showed in new supplementary Figure S7, S9 and S10). We acknowledge that this hypothesis is just speculative at this stage and need to be consolidated by further analyses/data/campaigns and for this reason we rephrased the paragraph as follows:

"These signals have never been observed before in ambient aerosol samples and are largely missing in the Signy samples. They can be tentatively attributed to acidic sugars (e.g., uronic acids) or organic sulfate (sulfate-esters), as better discussed in Supplementary (Figure S10 and corresponding text). Considering the high abundance of nSS-SO4 and the likely corresponding acidic nature of the aerosol in Halley, a hypothesis for the formation of these compounds can be the esterification of common polyols (such as glycerol) to organic sulfates. However, this hypothesis is just speculative at this stage and possibly needs confirmation from additional analysis/data."

About the interpretation of Factor 5 based on its spectral profile and about its difference with respect to Factor 1 and 2 we have already commented in the text (showing also some details in Figure S7) and we added in the revised version a more detailed description of factor analysis results (as mentioned above) showing even better our motivations.

340: if is not clear to me what the authors mean by "the changing position of the atmospheric polar front? Do you mean the change in latitude? There are seasonal shifts of the atmospheric polar front but the campaign described in Humphries et al is too short for those changes. There are day-to-day variations in strength and slight variations in position of the atm. polar front, but Humphries et al do not investigate that, it only reports a noticeable change in atmospheric composition upon crossing the atm. polar front (i.e. crossing into the polar cell).
Response: The Referee is right. We have modified the sentence into:
"Humphries et al. (2021) evidenced, in the area of East-Antarctica, latitudinal gradients in atmospheric aerosol loading and composition which were put in relation with the position of the atmospheric polar front"

350: based on what has it been determined that 60% of the air masses have been linked to katabatic winds? Continental wind direction is not sufficient to classify winds as katabatic winds. If katabatic winds are going to be discussed here, then I would suggest to define them.
Response: because we don't have the data to demonstrate the occurrence of katabatic outflows, we removed the definition and discuss only of "free-tropospheric circulation" over the continent.

380: instead of "across latitudes" please put at both locations
Response: we respectfully disagree with the Reviewer: here we want to stress exactly that Factor 3 is found in similar concentrations at both the sites because it is a background component, spread across a wide range of latitudes (from 60°S of Signy to 75°S of Halley and more, considering the circulation of the air through the Antarctic continent).

389: can you please clarify what do you mean by breaking barriers? It is very reasonable to expect factor 3 to be present at both locations, regardless of atmospheric circulation. I suggest to avoid using the term "across latitudes" as it is suggesting that measurements were done at many different locations. Is it WSOA or WSOM?
Response: we rephrased the sentence here as follows: "Nevertheless, we show that the secondary components associated with Factor 3 (MSA and DMA) can cross such gradients and be distributed at different latitudes."
WSOA is correct for us.

398-400: as mentioned previously not every Antarctic continental flow is katabatic flow.
Response: changed.

425: MSA should be reported separately and then report fractions of WSOM.
Response: done, as explained above and showed in the new figures.

440: This study had a relatively low number of samples (especially at Halley) so it seems a bit far-fetched to make conclusions about "chemical segregation" and prevention for certain OA types appearing at the site
Response: even if the number of samples is limited, the sampling period actually cover more than 1 continuous month in parallel at the two sites. We do not claim that our conclusions can be generalized to all Antarctica and/or all the seasons and we acknowledge they need to be confirmed by other studies in other sites and/or other periods at the same sites. Nevertheless, we report in the conclusions our best interpretation of the chemical composition we found in our study, the first going in such details and still not contradicted by any other (at least to our best knowledge).

700: Figure 2: Please split WSOM into MSA and the rest of WSOM (e.g. MSA lighter green; the rest darker green).
Response: done in the revised figure.

Figure 9: why does Signy (whole) not have error bars included?
Response: as already explained in a previous reply, error bars in the figures (where reported) represent the inter-sample variability and are showed only if useful to transmit the message and not confusing it. In figure 9 we have considered appropriate error bars only for reporting the statistics for parallel sample sets (either Signy and Halley), avoiding to add misleading very large bars for the whole Signy dataset (represented by the dashed histograms) because they should include the variability of two very different sampling periods with very different concentrations/features (as demonstrated before) making the figure hardly readable.

Supplement:
Line 30: for consistency, please use Positive Matrix Factorisation (PMF) throughout the text.
Response: agreed.

Additional references
Arrindell, W. A., & van der Ende. J.: An empirical test of the utility of the observations-to-variables ratio in factor and components analysis. Applied Psychological Measurement, 9, 165 – 178, 1985.

Baccarini, A., Dommen, J., Lehtipalo, K., Henning, S., Modini, R. L., Gysel-Beer, M., Baltensperger, U., and Schmale, J.: Low-volatility vapors and new particle formation over the Southern Ocean during the Antarctic Circumnavigation Expedition, J. Geophys. Res.-Atmos., 126, e2021JD035126, https://doi.org/10.1029/2021JD035126, 2021.

Brege, M., Paglione, M., Gilardoni, S., Decesari, S., Facchini, M. C., and Mazzoleni, L. R.: Molecular insights on aging and aqueous-phase processing from ambient biomass burning emissions-influenced Po Valley fog and aerosol, Atmos. Chem. Phys., 18, 13197–13214, https://doi.org/10.5194/acp-18-13197-2018, 2018.

Chalbot, M.C., and Kavouras, I.G.: Nuclear magnetic resonance spectroscopy for determining the functional content of organic aerosols: a review; Environ Pollut. 2014, Aug:191:232-49. doi: 10.1016/j.envpol.2014.04.034.

Cleveland, M.J., L.D. Ziemba, R.J. Griffin, J.E. Dibb, C.H. Anderson, B. Lefer, B. Rappenglück Characterisation of urban aerosol using aerosol mass spectrometry and proton nuclear magnetic resonance spectroscopy, Atmos. Environ., 54 (2012), pp. 511-518, 10.1016/j.atmosenv.2012.02.074

Costello, A. B., & Osborne, J. W.: Best practices in exploratory factor analysis: Four recommendations for getting the most from your analysis. Practical Assessment Research & Evaluation, 10(7), 2005. from http://pareonline.net/pdf/v10n7a.pdf

Decesari, S., Paglione, M., Mazzanti, A., Tagliavini, E.: NMR spectroscopic applications to atmospheric organic aerosol analysis – Part 1: A critical review of data source and analysis, potentialities and limitations, Trends in Analytical Chemistry (TrAC),Vol 171, 117516, https://doi.org/10.1016/j.trac.2023.117516, 2024.

Gilardoni, S., Massoli, P., Paglione, M., Giulianelli, L., Carbone, C., Rinaldi, M., Decesari, S., Sandrini, S., Costabile, F., Gobbi, G. P., Pietrogrande, M. C., Visentin, M., Scotto, F., Fuzzi, S., and Facchini, M. C.: Direct observation of aqueous secondary organic aerosol from biomass burning emissions, P. Natl. Acad. Sci. USA, 113, 10013–10018, 2016.

Graham B, Mayol-Bracero OL, Guyon P, Roberts GC, Decesari S, Facchini MC, Artaxo P, Maenhaut W, Koll P, Andreae MO (2002) J Geophys Res 107(D20):8047

Hogarty, K. Y., Hines, C. V., Kromrey, J. D., Ferron, J. M., & Mumford K. R.: The quality of factor solutions in exploratory factor analysis: The influence of sample size, communality, and overdetermination. Educational and Psychological Measurement, 65, 202-226, 2005.

Fridlind, A.M., and Jacobson, M.Z.: A study of gas-aerosol equilibrium and aerosol pH in the remote marine boundary layer during the First Aerosol Characterization Experiment (ACE1); Journal of Geophysical Research, 105, D13, 17325-17340, 2000.

Keen, W.C., Pszenny, A.A.P., Maben, J.R., Stevenson, E., and Wall, A.: Closure evaluation of size-resolved aerosol pH in the New England coastal atmosphere during summer; Journal of Geophysical Research, 109, D23307, doi:10.1029/2004JD004801, 2004.

Liu, J., Dedrick, J., Russell, L. M., Senum, G. I., Uin, J., Kuang, C., Springston, S. R., Leaitch, W. R., Aiken, A. C., and Lubin, D.: High summertime aerosol organic functional group concentrations from marine and seabird sources at Ross Island, Antarctica, during AWARE, Atmos. Chem. Phys., 18, 8571– 8587, https://doi.org/10.5194/acp-18-8571-2018, 2018.

MacCallum, R. C., Widaman, K. F., Preacher, K. J., & Hong S.: Sample size in factor analysis: The role of model error. Multivariate Behavioral Research, 36, 611-637, 2001.

MacCallum, R. C., Widaman, K. F., Zhang, S., & Hong S.: Sample size in factor analysis. Psychological Methods, 4, 84-99, 1999.

Preacher, K. J., & MacCallum, R. C.: Exploratory Factor Analysis in Behavior Genetics Research: Factor Recovery with Small Sample Sizes. Behavior Genetics, 32, 153-161, 2002.

Saiz-Lopez, A., Mahajan, A. S., Salmon, R. A., Bauguitte, S. J.-B., Jones, A. E., Roscoe, H. K., and Plane, J. M. C.: Boundary layer halogens in coastal Antarctica, Science, 317, 348, doi:10.1126/science.1141408, 2007.

Schkolnik, G., Rudich, Y. Detection and quantification of levoglucosan in atmospheric aerosols: a review. Anal Bioanal Chem 385, 26–33 (2006). https://doi.org/10.1007/s00216-005-0168-5

Slowik, J. G., Vlasenko, A., McGuire, M., Evans, G. J., and Abbatt, J. P. D.: Simultaneous factor analysis of organic particle and gas mass spectra: AMS and PTR-MS measurements at an urban site, Atmos. Chem. Phys., 10, 1969–1988, https://doi.org/10.5194/acp-10-1969-2010, 2010.

Suzuki, Y., Kawakami, M., and Akasaka, K.: 1H NMR Application for Characterizing Water-Soluble Organic Compounds in Urban Atmospheric Particles, Environ. Sci. Technol., 35, 2656–2664, https://doi.org/10.1021/es001861a, 2001.

Velicer, W. F., & Fava, J. L.: Effects of variable and subject sampling on factor pattern recovery. Psychological Methods, 3, 231-251, 1998.